# Single-cell transcriptomic profiling unveils dysregulation of cardiac progenitor cells and cardiomyocytes in a mouse model of maternal hyperglycemia

Sathiyanarayanan Manivannan[1,2,7], Corrin Mansfield [1,2,7], Xinmin Zhang[3], Karthik M. Kodigepalli[4], Uddalak Majumdar[1,2], Vidu Garg [1,2,5,6] & Madhumita Basu [1,2,5✉]

Congenital heart disease (CHD) is the most prevalent birth defect, often linked to genetic variations, environmental exposures, or combination of both. Epidemiological studies reveal that maternal pregestational diabetes is associated with ~5-fold higher risk of CHD in the offspring; however, the causal mechanisms affecting cardiac gene-regulatory-network (GRN) during early embryonic development remain poorly understood. In this study, we utilize an established murine model of pregestational diabetes to uncover the transcriptional responses in key cell-types of the developing heart exposed to maternal hyperglycemia (matHG). Here we show that matHG elicits diverse cellular responses in E9.5 and E11.5 embryonic hearts compared to non-diabetic hearts by single-cell RNA-sequencing. Through differential-gene-expression and cellular trajectory analyses, we identify perturbations in genes, predominantly affecting *Isl1*+ second heart field progenitors and *Tnnt2*+ cardiomyocytes with matHG. Using cell-fate mapping analysis in Isl1-lineage descendants, we demonstrate that matHG impairs cardiomyocyte differentiation and alters the expression of lineage-specifying cardiac genes. Finally, our work reveals matHG-mediated transcriptional changes in second heart field lineage that elevate CHD risk by perturbing *Isl1-GRN* during cardiomyocyte differentiation. Gene-environment interaction studies targeting the Isl1-GRN in cardiac progenitor cells will have a broader impact on understanding the mechanisms of matHG-induced risk of CHD associated with diabetic pregnancies.

[1] Center for Cardiovascular Research, Abigail Wexner Research Institute at Nationwide Children's Hospital, Columbus, OH, USA. [2] Heart Center, Nationwide Children's Hospital, Columbus, OH, USA. [3] BioInfoRx Inc., Madison, WI, USA. [4] Department of Pediatrics, Medical College of Wisconsin, Milwaukee, WI, USA. [5] Department of Pediatrics, The Ohio State University College of Medicine, Columbus, OH, USA. [6] Department of Molecular Genetics, The Ohio State University, Columbus, OH, USA. [7] These authors contributed equally: Sathiyanarayanan Manivannan, Corrin Mansfield. ✉email: Madhumita.Basu@ nationwidechildrens.org

Congenital heart disease (CHD) is the most common developmental malformation in humans and the leading cause of birth-defect related infant mortality[1,2]. CHD has multifactorial etiology, and among this heterogeneous category of developmental defects, some are the product of a gene–environmental interaction[1,3–6]. A disease-causing genetic abnormality is identified in ~20–30% of all CHD cases[7–10], however, the identification of genetic factors does not inform on the phenotypic variability among CHD patients with identical genetic variants. Numerous environmental factors, such as maternal pre-existing illnesses, viral infections, and therapeutic and nontherapeutic drug exposures have been identified to elevate the risk of CHD[6,11–14]. A strong correlation has been established between maternal pregestational diabetes mellitus (matPGDM) and an increased occurrence of CHD in the infants of diabetic mothers. In a study published in the *American Journal of Preventive Medicine*, the researchers found that uncontrolled blood sugar in women with type 1 or type 2 diabetes before pregnancy led to ~2,670 babies with CHD each year[15,16]. Therefore, it remains critical to understand the molecular responses to maternal hyperglycemia (matHG), a primary teratogen in matPGDM which alters CHD-risk genes to increase the frequency of the disease. Additional studies are necessary to explain the gene-environmental basis of cardiac lesions in patients harboring identical genetic variants.

During cardiogenesis, four precursor populations have been identified that contribute to different myocyte and nonmyocyte heart cell lineages: first heart field (FHF), second heart field (SHF), proepicardium, and cardiac neural crest (NC) cells[17,18]. Understanding the genes involved in each lineage is therefore essential for uncovering the environmental etiology of CHD[17,19]. Epidemiological studies reveal that matPGDM increases risks of most CHD phenotypes ranging from conotruncal and cardiac septal defects to hypoplastic left heart syndrome[16,20,21]. A Danish nationwide cohort study reported that the relative risk of cardiac defects originating from the anterior SHF (truncus arteriosus, tetralogy of Fallot, double-outlet right ventricle or DORV, left ventricular outflow tract obstruction, ventricular septal defect) is significantly higher from those lesions originated from the posterior SHF[16]. Studies from our group and others using a mouse model of diabetic embryopathy have recapitulated the occurrence of human CHD phenotypes, which further established the importance of gene-environment interaction[22–24]. However, relatively little is known about the genetic and molecular basis of these cardiac abnormalities (especially those with SHF origin) upon intrauterine exposure to matHG. In normal development, as the heart tube undergoes rightward looping during embryonic days (E)8.5–E10.5, the myocardium from SHF adds to the lengthening outflow tract (OFT) and further ballooning of future chambers takes place[25–27]. The SHF or pharyngeal mesodermal progenitor cells are situated medial to the FHF and are distinguished by the expression of genes encoding the transcription factors (TFs) Isl1, Mef2c, Foxh1, Foxc1/c2, Hand2, Symd1 (Bop), and growth factors Fgf8 and Fgf10[28,29]. Mouse knockouts targeting several of these TFs show early lethality by E10.5 due to failed looping, profound vascular defects, absent or single hypoplastic ventricular chamber, and defects in the coronary vessel and epicardial development[30,31]. Cre-lineage tracing and retrospective clonal analysis in avian and mouse embryos have demonstrated that SHF is multipotent and gives rise to the OFT, right ventricle (RV), and atrial cardiomyocytes (CM)[28,32,33] with additional contributions to the smooth muscle cells (SMC) and endocardial/endothelial cells (EC). The SHF specification and differentiation to CM is a tightly regulated process[28,34]. However, matHG induced transcriptional changes in SHF progenitors and their descendants underlying the elevated risk of CHD are not completely understood. Direct or indirect effects on the gene-regulatory program driven by TFs (Isl1, Tbx1, Prdm1, Six1, Nkx2.5, Gata4, Mef2c, Hand2), intracellular signaling pathways (Bmp, Fgf, Shh, Wnt, Notch), and chromatin remodeling factors (Smarca4, Smarcd3, Smarcc1) affect SHF deployment and result in a higher incidence of CHD[29,30,33]. We and others have demonstrated that exposure to matHG is a disruptor of gene-expression within the signaling pathways, and epigenetic modifiers, including Notch, Wnt, Bmp, Tgfb, Vegf, Shh, Hif1α, and Jarid2[22,24,35,36]. Gene-environment interaction studies between matPGDM and *Notch1*, *Nkx2.5*, *Ask1*, and *Hif1α* haploinsufficiency further revealed an increased occurrence of CHD in matHG-exposed embryos compared to controls[22,24,36–38]. While these studies highlight the need to dissect the molecular mechanism(s) of matHG exposure during cardiac development, the effect of matHG-mediated transcriptional changes in diverse cardiac cell lineages remain unknown. This impedes precise understanding of the developmental toxicity elicited by matHG in utero.

Here, we used the streptozotocin-induced (STZ) murine model of matPGDM to study the impact of matHG on cellular and molecular changes in developing embryonic hearts post looping morphogenesis. We found direct evidence of altered gene expression upon matHG exposure using single-cell RNA-sequencing (scRNA-seq) analysis between normoglycemic control (CNTRL) and matHG-exposed E9.5 and E11.5 whole hearts. Differential gene expression and functional enrichment analyses revealed HG-responsive changes in $Isl1^+$ multipotent SHF progenitors at E9.5, and in $Tnnt2^+$ CMs at E9.5 and E11.5. Further, by in vivo SHF cell-fate mapping studies in CNTRL and matHG-exposed $Isl1\text{-}Cre^+$; $Rosa^{mT/mG}$ E9.5, E11.5, and E13.5 hearts, we demonstrated CM differentiation defects when exposed to matHG environment. The combination of scRNA-seq data and in vivo validation of gene expression in Isl1+SHF-derived cells have revealed that matHG leads to dysregulated expression of cell lineage specifying TFs (Isl1, Tbx1, Tbx20, Fgf10, Mef2c, Nkx2-5, and Hand2) in multipotent progenitor cells. Furthermore, reduced CM proliferation and perturbations in the Isl1-dependent gene regulatory network (Isl1-GRN) in matHG-exposed embryos suggest underlying risk of CHD. Overall, this study delineates the impact of matHG on CM dysregulation and scRNA-seq data prioritizes on the cardiac progenitor cells as a major contributor to matPGDM-induced CHD.

## Results

**Single cell RNA-sequencing identifies widespread transcriptional dysregulation in E9.5 and E11.5 hearts in response to matHG.** To understand the cellular basis of matHG-induced risk of CHD and compare the cardiac cell-type-specific transcriptional responses in CNTRL and matHG-exposed embryonic hearts, we applied in vivo 10XscRNA-seq. We used a well-established murine matPGDM model that exhibits similar pathogenesis to human type 1 diabetes mellitus[24,39], in which we and others have demonstrated increased incidence of septal defects, DORV, and truncus arteriosus in matHG-exposed embryos[22,24,36–38]. For scRNA-seq experiment, embryonic hearts from one litter were harvested at E9.5 and E11.5 per condition to examine the effect of matHG on critical stages of cardiac development post looping morphogenesis and segmentation (Fig. 1a). Whole hearts were microdissected and pooled from at least six embryos at each timepoint and maternal hyperglycemic status (maternal B.G. = 145 mg/dl; $n = 8$ E9.5 embryos and 196 mg/dl; $n = 7$ E11.5 embryos used as untreated or CNTRL group and maternal B.G. = 312 mg/dl; $n = 6$ E9.5 embryos and 316 mg/dl; $n = 7$ E11.5 embryos used as experimental or matHG group). Cardiac tissues

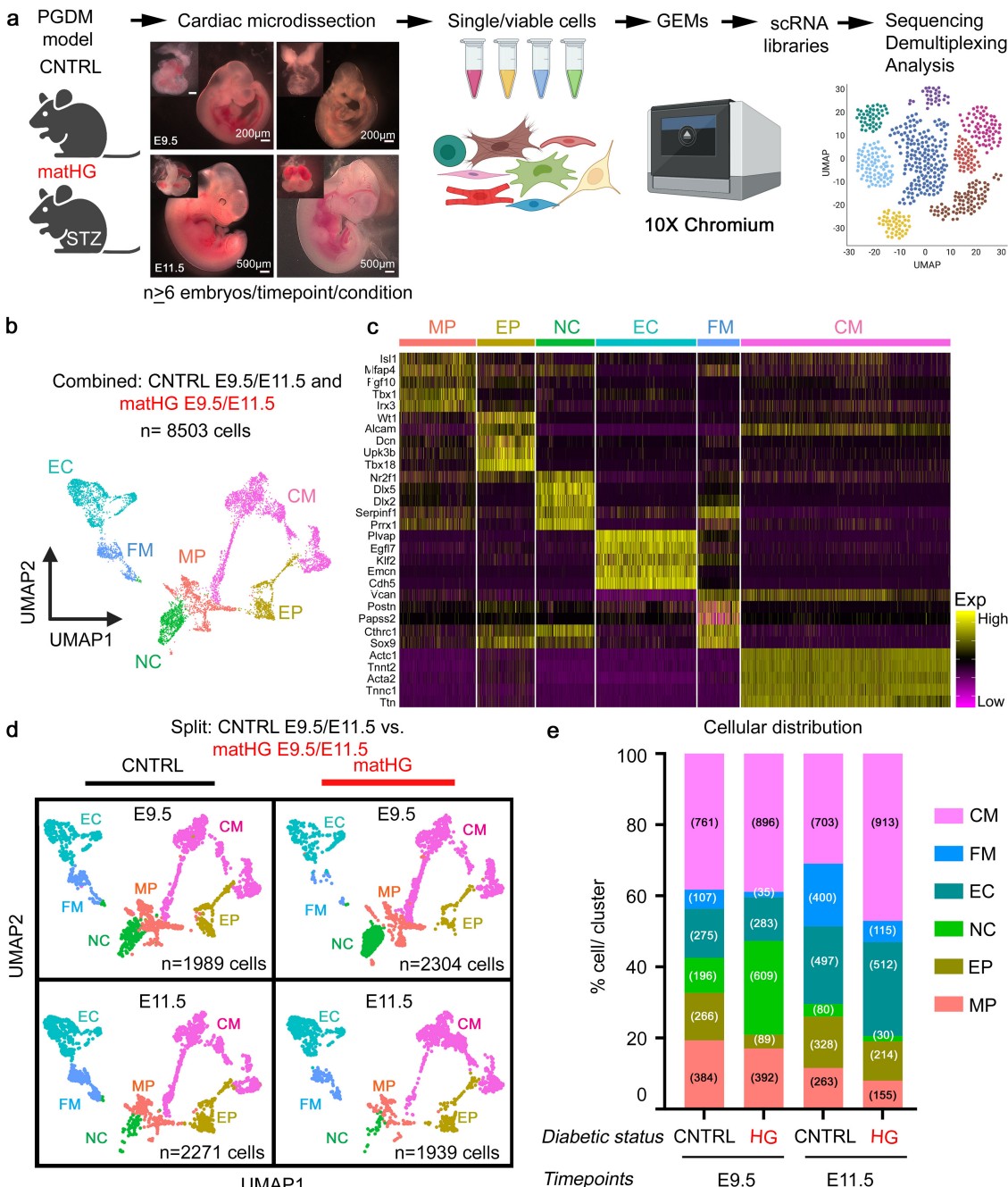

**Fig. 1 Single-cell transcriptomic sequencing reveals cell type-specific response to maternal hyperglycemia in E9.5 and E11.5 hearts. a** Workflow of the single-cell RNA-sequencing, created with BioRender.com. A representative image of E9.5 and E11.5 embryos demonstrate the cardiac regions dissected, and the pipeline for single-cell and 10X library preparation and next-generation sequencing At least six independent CNTRL and matHG-exposed embryos per embryonic timepoints were pooled and used for scRNA-seq analysis. Dissected hearts are shown in the insets. Scale bars: 200 and 500 μm. **b** UMAP plot of 8503 single-cell transcriptomes derived from CNTRL and matHG-exposed E9.5 and E11.5 hearts. Unsupervised clustering of cells with similar transcriptional profiles was clustered into six distinct cardiac cell populations, designated as MP, EP, NC, EC, FM, and CM. Each dot represents an individual cell and is colored according to cluster identities. Statistical tests for differential gene expression were applied to 8503 cells. **c** Heatmap shows the expression of the top five marker genes in each cluster from E9.5 and E11.5 scRNA-seq data. The rows represent cells and are ordered by cell cluster identities and hierarchical clustering. Normalized log expression levels are shown in yellow (high expression) and purple (low expression). **d** UMAP plots split into CNTRL and matHG-exposed cells in each cluster at E9.5 and E11.5 hearts, colored according to the cluster identities. Total cells, n = 1989; CNTRL E9.5, n = 2271; CNTRL E11.5, n = 2304; matHG E9.5, n = 1939, matHG E11.5. Numbers denote single-cell transcriptomes used to compare differential gene expression analysis between two groups. **e** Barplot indicates the proportion of cells in each cluster (in percentages) normalized to the total number of cells per sample at E9.5 and E11.5 stages (number of cells are in parenthesis). Colors indicate cluster identities. CNTRL control, matHG, maternal hyperglycemia, UMAP Uniform Manifold Approximation and Projection, MP multipotent progenitor, EP epicardial, NC neural crest, EC endocardial/endothelial, FM fibro-mesenchymal, CM cardiomyocytes.

from each developmental stage were dissociated and processed to obtain single-cell libraries using the 10xGenomics Chromium controller and 3′ polyA-based gene expression analysis kit followed by sequencing (Fig. 1a). Recently developed R Shiny apps 'Natian' and 'Ryabhatta' were used for pre-processing and scRNA-seq data analysis[40,41] as described in Supplementary Fig. 1. We captured transcriptomic data from 3042 and 4632 single cells from CNTRL embryos and 4022 and 3785 single cells from matHG-exposed E9.5 and E11.5 hearts, respectively. To classify the cardiac cell clusters, Seurat-based unsupervised clustering and Uniform Manifold Approximation and Projection (UMAP) based dimensionality reduction was performed after combining four samples at each developmental stage (Supplementary Fig. 2a–d and Supplementary Fig. 3a–f). Cells that passed the quality control (QC) metrics including the number of genes detected in each cell (nFeatureRNA), unique molecular identifiers (UMI or nCountRNA), and ≤10% of the reads mapped to mitochondrial genes, were used to analyze differentially expressed genes (DEGs) (Supplementary Fig. 2a–d and Supplementary Fig. 3a–f). The heatmap and UMAP distribution showed that after combining four samples, we identified 14 clusters (C0-C13) (Supplementary Fig. 2b, c), annotated based on the top five cell-type-specific marker genes obtained from our data and published scRNA-seq datasets of *wt* CNTRL embryonic hearts[42,43]. Clusters representing C3, C6, C7, C11, C12, and C13 expressed endoderm, ectoderm, blood/hematoendothelial markers and therefore were removed from further analysis (Supplementary Fig. 2c, e–g). In addition, we further extended the analysis by integrating our data with *wt* CNTRL E9.25 mouse heart cells reported by de Soysa, et al. (GSE126128)[43]. This integration showed that the cell types we annotated using cluster markers identified in our data closely track the cell types annotated by de Soysa, et al. (Supplementary Fig. 2h). Moreover, clusters that we removed from our analysis (did not have a pair in de Soysa, et al.[43]). These clusters include ectodermal markers, low UMI counts, and immune and blood cell markers. We note that similar clusters were also identified by de Soysa, et al., (GSE126128; Extended Figure data 1[43]) but excluded from further analysis to study early cardiac development. We applied a similar approach as described by de Soysa, et al. at each stage of development, which identified nine clusters with the heatmap showing the top five marker genes per cluster (Supplementary Fig. 3g, h). We also performed a power analysis on the single-cell data using two separate methods. Using SCOPIT[44], for prospective power analysis, we showed that we could detect a cell type with a frequency of 1% in the observed population, with the number of cells captured per sample per time point with high confidence (Supplementary Fig. 4a–c). Also, using SCOPIT retrospective analysis, we showed that the cell types with the lowest frequency in each sample can be detected in other samples with high confidence using the number of cells captured by our single-cell data. This suggests that our sample size (in terms of the number of cells) was sufficient to capture every cell type observed in control E9.5 data across test conditions (Supplementary Fig. 4a–c). This also means that we controlled for sampling bias when evaluating cellular proportions. While the individual embryos were not barcoded a priori, we used post hoc sequencing analysis to infer the genotypes of the embryos using a genotype-free demultiplexing tool "souporcell"[45], which allowed us to infer the genotypes of at least 6 embryos per sample (Supplementary Fig. 5a–e).

Following the QC steps, the single-cell transcriptomes from a total of 8503 cells were classified into six broadly defined cardiac populations that were used to compare cell-type-specific DEGs between CNTRL and matHG-exposed E9.5 (1989 and 2304 cells, respectively) and E11.5 (2271 and 1939 cells, respectively) hearts

(Fig. 1b–d). These clusters represented *Isl1*+ and *Tbx1*+ multipotent progenitors (MP), *Tnnt2*+ *and Actc1*+ cardiomyocytes (CM), *Cdh5*+ and *Emcn*+ endocardial/endothelial (EC), *Postn*+ and *Sox9*+ fibro-mesenchymal (FM), *Wt1*+ and *Tbx18*+ epicardial (EP), and *Dlx2*+ *and Dlx5*+ neural crest (NC) cells. Clustering annotation was performed by finding the gene expression signature of each cluster using marker genes that delineate cell identities as described in published scRNA-seq datasets (Supplementary Fig. 2h) from *wt* embryonic hearts[41–43]. The UMAP and dot plots showed cluster-specific expression of marker genes used to classify the above cell types (Supplementary Fig. 6a, b).

Next, we compared the proportion of cell clusters per embryonic timepoint between CNTRL and matHG-exposed E9.5 and E11.5 hearts (Fig. 1e and Supplementary Fig. 7a, d). CM were the most abundant cell type at both time points, yet only significantly elevated in matHG-exposed hearts compared to CNTRL at E11.5 (47.1% vs. 30.9%, $p < 0.0001$). The MP cell population in matHG hearts was significantly decreased compared to CNTRL at E9.5 (17.0% vs. 19.3%, $p = 0.05$) and E11.5 (7.9% vs. 11.6%, $p < 0.0001$). Similarly, the FM cells were significantly lower in the E9.5 (1.5% vs. 5.4%, $p < 0.0001$) and E11.5 (5.9% vs. 17.6%, $p < 0.0001$) hearts subjected to matHG compared to CNTRL hearts. In contrast, a significant increase in EC cells in response to matHG exposure only at E11.5 (26.4% vs. 21.9%, $p = 0.0007$). The proportion of EP cells was significantly reduced at both E9.5 (3.9% vs. 13.4% $p < 0.0001$) and E11.5 (11.0% vs. 14.4%, $p = 0.001$) in matHG exposed hearts compared to CNTRL. NC-derived cells were significantly enriched in matHG-exposed E9.5 heart (26.4% vs. 9.8%, $p < 0.0001$) but significantly reduced (1.5% vs. 3.5%, $p < 0.0001$) by E11.5 stage (Fig. 1e and Supplementary Fig. 7a, d). These findings from in vivo scRNA-seq data reveal that intrauterine exposure of matHG induces diverse cellular responses and results in abnormal cellular distribution at the early stages of cardiac development.

**Hyperglycemia elicits transcriptional changes in multipotent cardiac progenitor cells and cardiomyocytes.** To gain a deeper understanding of how matHG exposure affects multiple signaling pathways and their intersection with transcriptional regulatory networks in CM lineage, we examined the transcriptional changes in MP and CM clusters from CNTRL and HG-exposed E9.5 and E11.5 scRNA-seq data. The proportion of MP cells showed significant differences between CNTRL and matHG-exposed E9.5 and E11.5 hearts, although CM were significantly altered only at E11.5 (Fig. 2a and Supplementary Fig. 7a, d). However, the normalized gene expression of *Isl1*+/*Tbx1*+/*Tnnt2*+ cells in combined MP-CM clusters may suggest the presence of less differentiated CM (Supplementary Fig. 8a, b). DESeq2 analysis in E9.5 *Isl1*+ MP cells (log$_2$FoldChange ≤ −1 or ≥ +1 and $P_{adj}$ ≤ 0.05) revealed 262 DEGs between CNTRL and matHG-exposed groups (Supplementary Data 1). Gene Ontology (GO) enrichment analysis of the DEGs revealed perturbations in genes associated with biological processes affecting (i) regionalization, anterior-posterior pattern specification, cell-fate commitment, (ii) cardiomyocyte, mesenchymal and neural crest cell differentiation, and (iii) cardiac ventricle and septum development (Supplementary Fig. 8c, Supplementary Data 2). DEGs associated with these processes include Hox-family members, fibroblast growth factors, Forkhead box members, T-box TFs, muscle-specific TFs, and regulators of SHF development (Supplementary Data 1); representative DEGs are shown in bubble plots (Supplementary Fig. 9a). Hence, DEG analysis of E9.5 *Is1l*+ MP cells revealed disruption of cardiac progenitor cell commitment genes associated with altered CM fate in response to matHG.

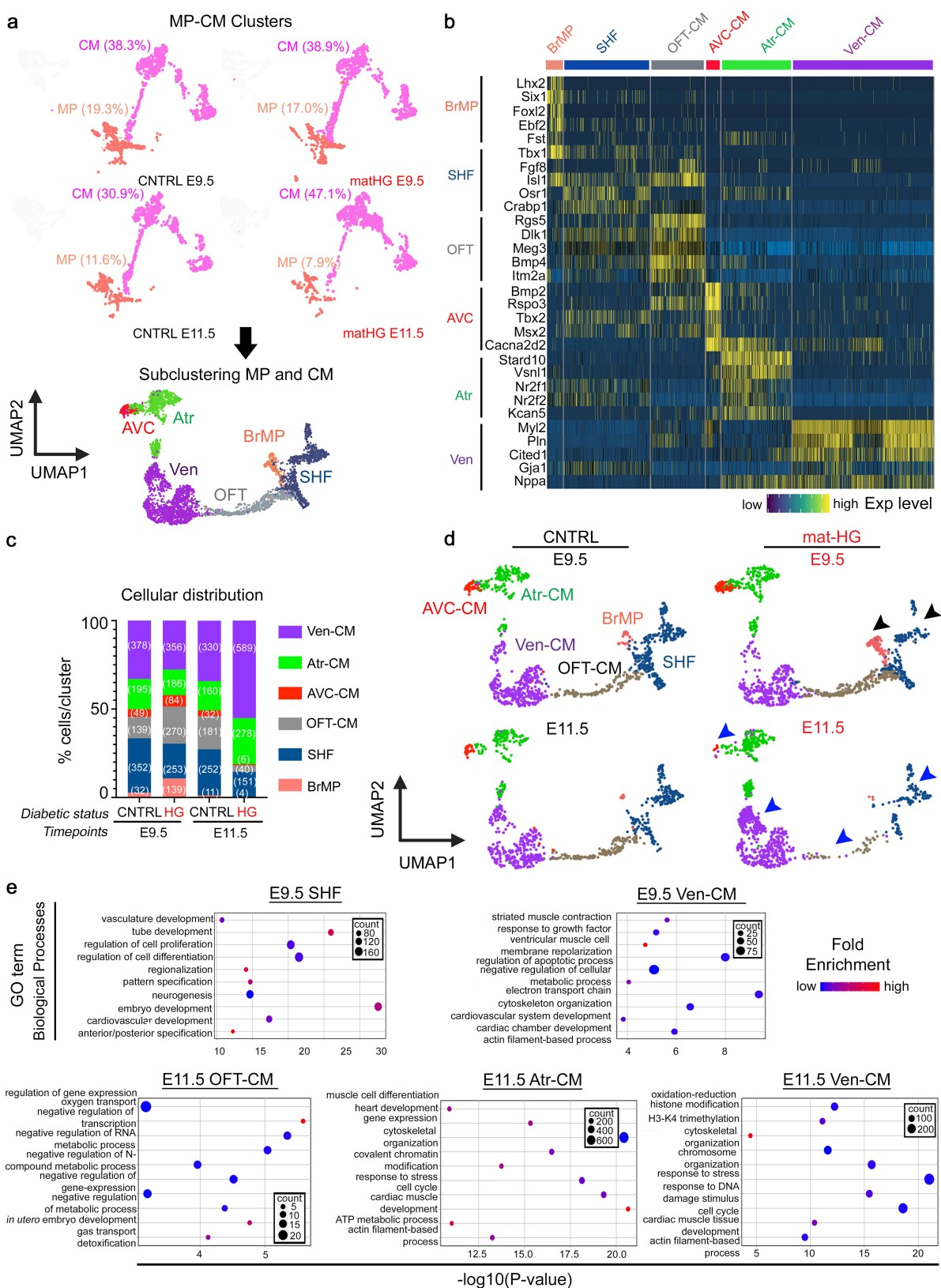

Next, differential expression analysis was performed in *Tnnt2*⁺ CM between CNTRL and matHG-exposed hearts at E9.5 and E11.5, which revealed 357 and 326 DEGs, respectively (Supplementary Data 1 and 3). GO analysis of CNTRL vs. matHG-exposed E9.5 and E11.5 CM showed molecular changes associated with (i) muscle contraction and myofibril assembly,

(ii) regulation of ion transport, (iii) cardiac muscle cell differentiation and proliferation, (iv) regulation of Wnt, Bmp, TGF and MAPK signaling pathways, (v) response to hypoxia, (vi) H3K4 trimethylation, and (vii) mitochondrial organization with the regulation of metabolic processes such as tricarboxylic acid cycle (Supplementary Fig. 8d, e and Supplementary Data 2, 4).

**Fig. 2 Marker Gene Expression profiling reveals differences in cardiac progenitor cells and cardiomyocyte subpopulations under matHG exposure.**
**a** UMAP plots represent the distribution of MP and CM clusters in CNTRL and matHG-exposed E9.5 and E11.5 hearts. Total MP cells, $n = 384$; CNTRL E9.5, $n = 263$; CNTRL E11.5, $n = 392$; matHG E9.5, $n = 155$, matHG E11.5. Total CM, $n = 761$; CNTRL E9.5, $n = 703$; CNTRL E11.5, $n = 896$; matHG E9.5, $n = 913$, matHG E11.5. The percentage of MP and CM is indicated in parenthesis. Unsupervised clustering of combined MP (1194 cells; 14.1%) and CM (3273 cells; 38.5%) show cells with similar transcriptional profiles and subclustered into six cellular subtypes. MP cells were clustered into SHF and BrMP and CM were clustered into OFT-CM, AVC-CM, Atr-CM, and Ven-CM subtypes based on marker gene analysis. Colors denote the identity of subclusters. Statistical tests for differential gene expression were applied to 4467 cells. **b** The heatmap shows the expression of the top five marker genes in each subcluster. The rows represent cells and are ordered by cell cluster identities and hierarchical clustering. Normalized log expression levels are shown in yellow (high expression) and blue (low expression). **c** Barplot indicates cellular distribution in each subcluster (in percentages) normalized to the total number of cells per sample at the E9.5 and E11.5 stages (number of cells is in parenthesis). Cluster identities are indicated in colors. **d** UMAP plots split into CNTRL and matHG-exposed MP and CM subpopulations at E9.5 and E11.5 hearts, colored according to the cluster identities. matHG-exposed SHF and BrMP cell populations at E9.5 and SHF, OFT, Atr, and Ven-CMs at E11.5 hearts are indicated by black and blue arrowheads, respectively. **e** Bubble plots represent enriched GO terms (biological process) in E9.5 SHF and Ven-CM and in E11.5 OFT, Atr and Ven-CM exposed to matHG. The colors of the nodes are illustrated from red to blue in descending order of $-\log 10$ (P value) and fold enrichment. The count represents gene number, also indicated by circle size, whereas the color denotes the up (red) or downregulation (blue) of the specific GO term in the cellular subpopulations. The horizontal (X) axis represents the $-\log 10$ (P value), and the vertical (Y) axis represents the GO terms. CNTRL control, HG hyperglycemia, MP multipotent progenitors, SHF second heart field, BrMP branchiomeric muscle progenitors, OFT outflow tract, AVC atrioventricular canal, Atr atrial, Ven ventricular, CM cardiomyocytes, GO gene ontology.

DEGs in E9.5 CM were primarily associated with cell differentiation, voltage-gated calcium channels, and potassium channels, regulators of cardiac contractility, transcriptional and chromatin regulators. Likewise, in E11.5 CM, expression of genes regulating the sarcomere assembly and cardiac muscle function, myocardial TFs, Bmp, TGF and EGF receptor family genes, glucose and mitochondrial metabolism were significantly perturbed with matHG (Supplementary Fig. 8e). Representative DEGs are shown in bubble plots (Supplementary Fig. 9b, c). Together, gene expression differences demonstrate that matHG exerts developmental toxicity on $Isl1^+$ MP and in $Tnnt2^+$ CM by affecting key biological processes. This data also suggests that dysregulated expression of genes important for CM lineage commitment is likely contributing to a spectrum of conotruncal defects observed in matHG-exposed fetuses.

**Heterogeneity in matHG-induced gene expression differences in MP and CM subtypes.** To determine the effects of matHG on transcriptional differences in MP and CM subpopulations, we performed a sub-clustering analysis on the integrated E9.5 and E11.5 scRNA-seq data (Fig. 2a). Unsupervised clustering of cells showed two subpopulations of MP and four subpopulations of CM present in the scRNA-seq data. Unsupervised clustering, dimensionality reduction, and heatmap analysis identified the marker gene expression in each subcluster. Based on the expression of the top five markers per cluster from previously published scRNA-seq data and overlap of *wt* E9.25 hearts[42,43], the MP cluster was further clustered into anterior/posterior SHF and branchiomeric muscle progenitor (BrMP) cells, whereas CM were subclustered as outflow tract (OFT), atrioventricular canal (AVC), atrial-CM (Atr-CM) and ventricular-CM (Ven-CM) (Fig. 2b). Marker gene expression for each CM subtype is shown in dot plots (Supplementary Fig. 10a). CNTRL and matHG-exposed E9.5 and E11.5 hearts showed significant differences in distribution of MP-CM subpopulations (Fig. 2c and Supplementary Fig. 7b, e). Upon matHG exposure, there was a significantly higher proportion of BrMP cells at E9.5 (10.8% vs. 2.8%, $p < 0.0001$); fewer BrMP cells were captured in E11.5 hearts, although not significantly different between treatments. There was a lower proportion of SHF populations in both E9.5 (19.6% vs. 30.7%, $p < 0.0001$) and E11.5 (14.1% vs. 26.1%, $p < 0.0001$) matHG-exposed hearts compared to controls. Given the contribution of MP cell subpopulations in heart development, the changes in proportions of progenitor subpopulations suggest a potential role of glucose sensitivity in cells from the SHF and

increased risk of CHD in the diabetic offspring[16,46,47]. This is further reflected in the significant reduction of $Tnnt2^+$ OFT-CM (3.7% vs. 18.7%, $p < 0.0001$) and AVC-CM (0.6% vs. 3.3%, $p < 0.0001$) at E11.5 with matHG (Supplementary Fig. 7b, e). We found a trend towards reduction of Atr-CM (14.4% vs. 17.0%; $p = 0.083$) and Ven-CM (27.6% vs. 33.0%; $p = 0.004$) at E9.5 upon matHG exposure compared to CNTRL, but were significantly higher by E11.5 (26.0% vs. 16.6%; $p < 0.0001$ and 55.1% vs. 34.2%; $p < 0.0001$ for Atr-CM and Ven-CM, respectively) when subjected to matHG environment (Fig. 2c, d and Supplementary Fig. 7b, e).

To characterize matHG-mediated transcriptional changes, we evaluated the expression of marker genes in CM subpopulations at two developmental timepoints (Supplementary Fig. 10b). DEG analysis revealed a trend towards downregulation of $Myl2^+$, $Gja1^+$, $Pln^+$ and $Cited1^+$ cells in Ven-CM, $Bmp4^+$ and $Itm2a^+$ cells in OFT-CM, $Bmp2^+$, $Rspo3^+$ and $Tbx3^+$ cells in AVC-CM and $Nr2f1^+$ Atr-CM at E9.5 (Supplementary Fig. 10b, Supplementary Data 5). In E11.5 hearts, Ven-CM expression of $Gja1^+$, OFT-CM expression of $Bmp4^+$, $Rgs5^+$ and $Dlk1^+$, and Atr-CM expression of $Cacna2d2^+$, $Tbx3^+$ AVC-CM and $Kcan5^+$, $Nr2f1^+$, $Nr2f2^+$ were significantly reduced with matHG. In contrast, there was significant upregulation of Atr-CM expression of $Ankrd1^+$, $Tubb5^+$, $Myl9^+$, $Myl1^+$ and Ven-CM expression of $Tmsb10^+$, $Pfn1^+$, $Prdx1^+$, $Cfl1^+$ by E11.5 (Supplementary Data 6). MatHG-driven dysregulation of TFs essential for SHF deployment and genes encoding CM sarcomeric proteins[48–55] suggest a plausible underlying mechanism linking transcriptional changes to diabetes-induced CHD and CM hypertrophy. This data also suggests that elevated glucose levels during pregnancy might affect fetal CM maturation by altering the expression of genes essential for the sarcomeric organization.

Next, GO analysis was performed on MP-CM DEGs and significantly enriched biological processes (FDR adjusted $p$ value $\leq 0.05$) were compared between maternal glycemic status in two developmental stages (Fig. 2e, Supplementary Data 5–7). DEGs in E9.5 SHF-derived cells were enriched in biological processes related to cell proliferation/differentiation; and anterior-posterior pattern specification, as described earlier in Supplementary Fig. 8c, whereas the DEGs in Ven-CM were associated with chamber development, muscle cell membrane polarization, and muscle contraction. The OFT-CM, Atr-CM, and Ven-CM at E11.5 hearts showed significant differences in gene expression related to stress response, muscle cell differentiation, muscle contraction, ATP-dependent metabolic processes, chromatin

modification and cell cycle genes (Fig. 2e, Supplementary Data 5–7). Thus, DEG and GO enrichment analyses of MP-CM subtypes showed that matHG disrupts the critical stages of heart development by affecting genes related to pattern specification and CM function.

**Transcriptomic analysis reveals the effect of matHG exposure on SHF-CM lineage.** To investigate the impact of matHG on transcriptional changes in MP-CM cell lineages, we performed pseudotime ordering of CNTRL and matHG-exposed E9.5 and E11.5 cells using both Monocle (version 2.0)[56] and Slingshot (version 1.8.0)[57] independently. Pseudotime analysis using Monocle2 revealed five distinct cell states, identified as States 1–5 (Fig. 3a). Overlaying the cluster identities with pseudotime trajectory revealed the distribution of SHF, BrMP, OFT-CM, AVC-CM, Atr-CM, and Ven-CM in each state. The cell composition suggested that States 1 and 5 are more progenitor-like, comprised of 70.8% SHF, 10.9% BrMP, 18.0% OFT-CM, 0.1% Atr-CM, and 0.2% Ven-CM (in State 1) and 50% SHF, 34.8% BrMP, 14.3% OFT-CM, and 0.9% Atr-CM (in State 5). State 2 suggested an intermediate state for differentiating CM containing 6.2% SHF, 0.5% BrMP, 64.1% OFT-CM, 6.7% Atr-CM, and 22.6% Ven-CM. While cell type distribution in States 3 and 4 suggested that these states are comprised of differentiated, more mature CM (Fig. 3a). State 3 was composed of 12.7% OFT-CM, 25.8% Atr-CM, 52.8% Ven-CM, and 8.7% AVC-CM, and State 4 had 1% SHF, 0.3% BrMP, 0.5% OFT-CM, 33.6% Atr-CM, 64.4% Ven-CM, and 0.2% AVC-CM. The percentages of cells/state were compared between CNTRL and matHG-exposed groups at each timepoint and statistical significance was determined using both chi-square and Fisher Exact tests (Fig. 3b and Supplementary Fig. 7c, f). At E9.5, we found significant differences in States 2, 3 and 5 with matHG, then changes in cellular distribution across the trajectory were shown to be significant in States 1–4 at E11.5 (Fig. 3b and Supplementary Fig. 7c, f). We then explored the pseudotime changes in this population along with the cell-cluster annotation from the Seurat analysis. We generated a scatter plot of the cells by plotting UMAP_1 reduction value from Seurat dimensionality reduction analysis against Monocle2-computed pseudotime[56]. Further, to capture the trajectories in this scatter plot, we used Slingshot to draw smoothened trajectories (Fig. 3c). Similar to DDRTree based reduction from Monocle2[56], Slingshot[57] trajectory analysis revealed progenitor-like States 1 and 5 (far left in the pseudotime), intermediate State 2 with less differentiated OFT-CM and Ven-CM and more differentiated Atr-CM and Ven-CM present at States 3 and 4 (far right in the pseudotime), superimposed with cluster identities as depicted in Fig. 3a. GO enrichment analysis between States 1 and 5 and States 3 and 4 identified DEGs associated with heart development, tube morphogenesis, metabolic processes, and cardiac muscle contraction, distinctly separating the progenitor cells and CM (Fig. 3d). Principal component analysis plots for CNTRL and matHG-exposed E9.5 and E11.5 are shown according to their true pseudotime, with trajectories inferred by Slingshot (Fig. 3c). This revealed differences in pseudotime trajectories at intermediate State 2 at E9.5 with matHG exposure, are likely due to an enriched population of less differentiated $Tnnt2^+$ CM or higher $Isl1^+$ expressing cells (Fig. 3c). Trajectory-based DEG analysis of States 1–5 was performed at each developmental timepoints (Supplementary Data 8, 9). We analyzed the top 20 GO terms (biological processes) for each pseudotime state obtained from GO enrichment analysis of DEGs. The enrichment analysis of DEGs between CNTRL vs. matHG exposed States 1 and 5 (progenitor-like) and States 3 and 4

(differentiated CM) revealed that matHG perturbed genes were linked to cardiac development, tube morphogenesis, cell differentiation, muscle structure development, and changes in oxidative phosphorylation and ATP-dependent metabolic processes (Fig. 3d, Supplementary Fig. 11a–e, and Supplementary Fig. 12a–e). In summary, these findings from pseudotime trajectory-based expression analysis reveal that matHG exposure sensitizes MP-CM lineage to affect CM differentiation.

**Fate mapping of SHF progenitors identifies impairments in CM differentiation under matHG exposure.** Marker gene analysis between CNTRL and matHG-exposed E9.5 and E11.5 hearts across the pseudotime trajectory revealed changes in $Isl1^+Tbx1^+$ MP (States 1 and 5) and $Tnnt2^+Actc1^+$ CM (States 2–4) (Fig. 4a). To examine the impact of matHG on the fate of SHF-derived CM differentiation, we quantified the expression of $Isl1$ and $Tnnt2$ along the pseudotime trajectory normalized to total number of cells from scRNA-seq data. A significantly higher percentage of $Isl1^+Tnnt2^+$ expressing cells in the less differentiated intermediate State 2 were noted in matHG-exposed E9.5 hearts (66/1288 cells; 5.12%) compared to CNTRL hearts (12/1145 cells; 1.05%, $p < 0.0001$) and a significant reduction in $Tnnt2^+$ CM (differentiated, State 3) (229/1068 cells; 21.4% in matHG vs. 484/966 cells; 50.1% in CNTRL, $p < 0.0001$) at E11.5 (Fig. 4a). This observation suggested that SHF-derived CM differentiation might be affected in response to matHG exposure.

To investigate if $Isl1$-derived CM differentiation is perturbed in vivo, we used $Isl1$-$Cre^{+/-}$ and $Rosa^{mT/mG}$ dual reporter mice for cell lineage tracing studies. CNTRL and STZ-treated matHG $Rosa^{mT/mG}$ females were bred with $Isl1$-$Cre^+$ males and E9.5, E11.5, and E13.5 embryos were collected to analyze the impact on CM differentiation (Fig. 4b). Average maternal B.G. level was significantly higher in matHG-exposed ($n = 11$, $471.3 \pm 182.9$ mg/dl) than CNTRL ($n = 9$, $217.7 \pm 27.7$ mg/dl) dams (two-tailed $p = 0.0007$) (Fig. 4c and Supplementary Table 1). Representative GFP expression in E9.5-E13.5 $Isl1$-$Cre^+$; $Rosa^{mTmG/+}$ embryos is shown in Supplementary Fig. 13a. The pattern of GFP expression mirrors previously described endogenous $Isl1$ expression in pharyngeal mesoderm, cardiac OFT, and foregut endoderm at E9.5 with extension to the regions of midbrain, forebrain, all cranial ganglia, spinal motor neurons, dorsal root ganglia, and in the posterior hindlimb of E11.5 and E13.5 $Cre^+$ embryos[58]. Subsequently, we performed co-immunostaining experiments in the E9.5-E13.5 transverse tissue sections with α-GFP (to mark $Isl1$-derived cells) and α-Tnnt2 (to trace $Isl1^+$SHF-derived CM) exposed to CNTRL and matHG environment. Cell-fate mapping revealed a significant reduction ($p = 0.001$) in the number of GFP$^+$Tnnt2$^+$ cells in matHG-exposed E9.5 hearts compared to CNTRL $Cre^+$ embryos (Fig. 4d, e, p). $Isl1$-driven expression of GFP reporter was similarly compared in E11.5 and E13.5 in CNTRL and matHG-exposed $Cre^+$ embryos. There was a significant reduction in Isl1-derived OFT-CM and Atr-CM of matHG exposed E11.5 hearts (both $p = 0.029$ and $0.007$) and a trend toward fewer in the RV (Fig. 4f–i, p). At E13.5, we found significant downregulation of GFP$^+$Tnnt2$^+$ cells in both the OFT and RV (Fig. 4j–p, both $p = 0.003$ and <0.0001). In addition to CM, Isl1$^+$ SHF cells also give rise to EC[59]. Therefore, we examined the effect of matHG on SHF-derived EC cells by evaluating $Emcn$ expression in scRNAseq data (Supplementary Fig. 13b, c). $Emcn$ mRNA expression was not significantly altered with matHG, however we detected very few $Isl1^+Emcn^+$ cells in situ at E9.5 and E11.5 hearts. Immunohistochemical analysis of the cardiac sections showed no discernable changes in GFP$^+$Emcn$^+$ cells marking SHF-derived EC in matHG-exposed $Is1l$-$Cre^+$; $Rosa^{mTmG/+}$ E9.5 and E11.5 embryos (Supplementary Fig. 13d–k). In conjunction with scRNA-seq, the in vivo cell

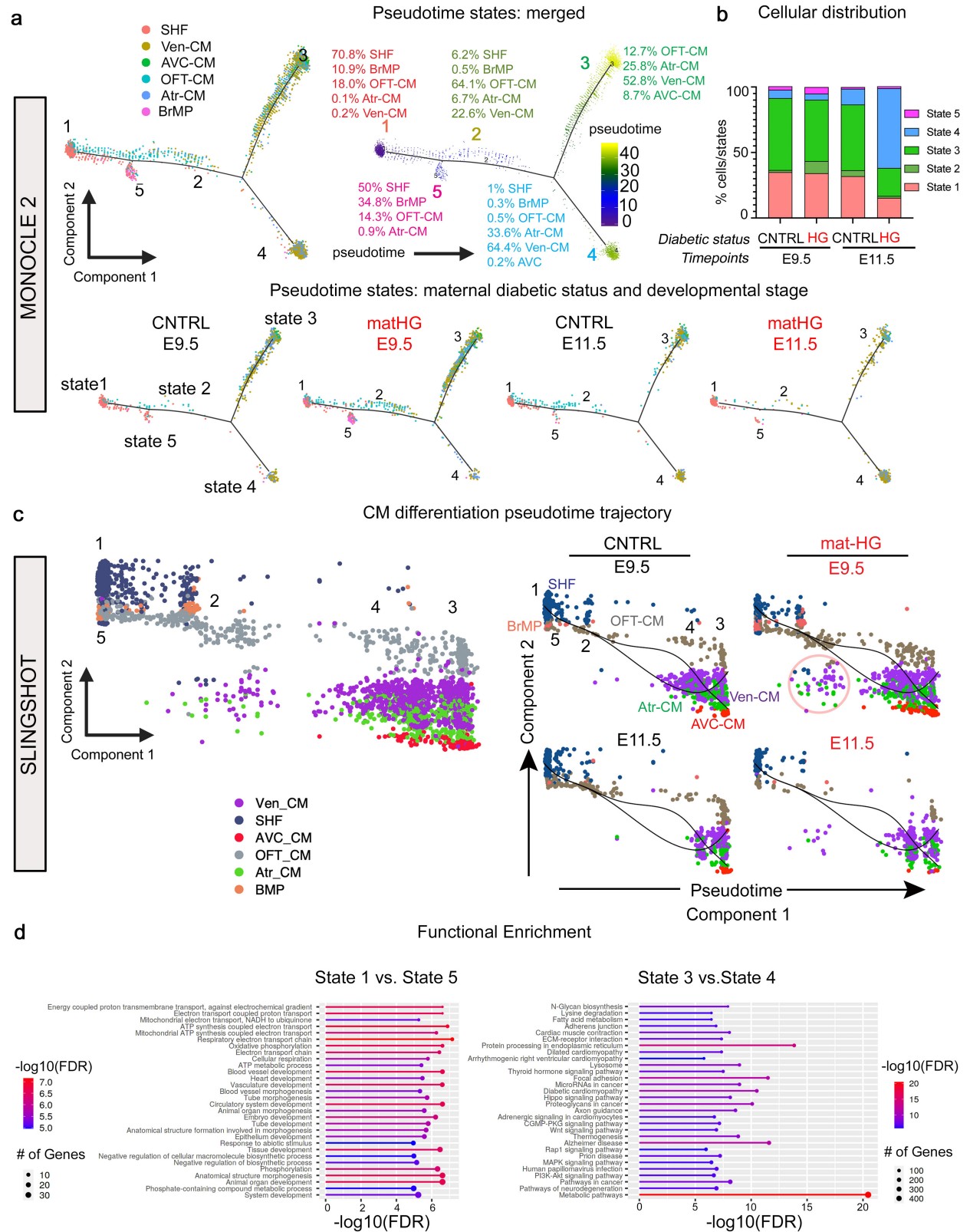

lineage tracing data demonstrates that the *Isl1*+-SHF-cells are sensitive to matHG environment resulting in impaired CM differentiation.

**Maternal hyperglycemia affects CM proliferation and lineage specifying genes**. The number of CM in the developing heart is determined by both the proliferation of the precursor multipotent

cells as well as the expansion of the differentiating CM[60]. We and others have previously demonstrated that matHG exposed embryos with septal and conotruncal defects display reduced CM proliferation at E13.5 as measured by the number of PHH3+ and BrdU+ cells[24,39]. Here, by scRNA-seq we showed that genes and pathways related to cell proliferation were also perturbed in E9.5 and E11.5 matHG-exposed CM (Supplementary Fig. 8d, e). To

**Fig. 3 Pseudotime ordering of MP-CM clusters reveals altered CM differentiation trajectory in response to matHG exposure. a** Pseudotime analysis of CNTRL and matHG-exposed E9.5 and E11.5 MP and CM subpopulations (SHF, BrMP, OFT, AVC, Atr, and Ven-CM) using Monocle 2. The merged and individual pseudotime trajectories are illustrated. Cells on the tree are colored according to cluster identities, state, and expression levels in pseudotime. The top 500 genes with the highest variability in expression were used to construct the pseudotime tree. The arrangement of cells shows the cells on the left side of the tree (dark blue) are less differentiated (progenitor-like) than those on the right side (bright yellow) and are more differentiated (CM subpopulations). Overlaying the cluster information shows that cells in states 1 and 5 correspond to the SHF and BrMP cells and states 2, 3, and 4 correspond to less and more differentiated CM, shown by percentages in each state. **b** Barplot indicates the proportion of cells in MP-CM subpopulations (in percentages normalized to total number of cells) across five pseudotime states (1–5) for CNTRL and HG-exposed E9.5 and E11.5 embryos. **c** Slingshot-based pseudotime trajectories calculated from UMAP-pseudotime embedding illustrate the trajectories of MP-CM subpopulations in CNTRL and matHG exposed E9.5 and E11.5 hearts (merged and split). State identities match the monocle states. Cells on the tree are colored according to cluster identities, state, and expression levels in pseudotime. The black line represents the developmental trajectory in pseudotime. The red circle in the matHG-exposed E9.5 trajectory indicates less differentiated CM at State 2, enriched by OFT-CM. **d** Functional enrichment analysis of differentially expressed genes between CNTRL vs. matHG exposed States 1 and 5 (progenitor-like) and States 3 and 4 (differentiated CM) using ShinyGO v0.741. Y-Axis indicates the pathway name or GO-term, X-axis indicates -log10(FDR) for the enriched pathways/terms. The bubble size indicates the number of genes. The color bar indicates the adjusted $p$ value (-log10(FDR), blue represents a higher value, and red represents a lower value. Statistical tests for differential gene expression were applied to 4467 cells. MP multipotent progenitor, CM cardiomyocytes, CNTRL control, HG hyperglycemia, BrMP branchiomeric muscle progenitors, OFT outflow tract, AVC atrioventricular canal, Atr atrial, Ven ventricular.

assess the impact of transcriptional changes of cell-cycle genes on Isl1⁺SHF-derived CM proliferation, we compared the percentage of mitotic cells between CNTRL and matHG-exposed hearts. We performed co-immunostaining with α-GFP and α-PHH3(Ser10), a mitosis maker in *Is1l-Cre⁺; Rosa^mTmG/+* E9.5 and E11.5 hearts and found a trend towards lower GFP⁺PHH3⁺ cells at E9.5 (Fig. 5a–d, i) and significant downregulation by E11.5 (Fig. 5e–i, $p = 0.072$ and 0.002) upon matHG exposure. Gene-expression-based cell cycle scoring analysis[61] of scRNAseq data also revealed a lower proportion of matHG-exposed CM are in the S/G2/M phases of the cell cycle compared to CNTRL embryos (Fig. 5j). This data was corroborated by the expression of cell-cycle genes (*Mki67, Ccnd2, and Ccnd1*) in the OFT-CM, AVC-CM, Atr-CM and Ven-CM (Supplementary Data 5 and 6).

Next, to examine the differential expression of SHF progenitor/CM differentiation markers obtained from scRNA-seq data, we performed candidate gene-based qRT-PCR in CNTRL and matHG-exposed *Is1l-Cre⁺; Rosa^mTmG/+* E9.5 and E11.5 hearts (Fig. 5k–o). GFP⁺ and GFP⁻ cells were obtained from *Cre⁺* hearts by fluorescence-activated cell sorting (FACS). Significant upregulation of GFP expression (83.6-136.9-fold) in CNTRL and matHG-exposed *Is1l-Cre⁺; Rosa^mTmG/+* E9.5/E11.5 hearts compared to GFP⁻ population from the same sample confirmed an effective isolation of two cell populations by FACS (Fig. 5p). Comparison between CNTRL and matHG-exposed E9.5 GFP⁺ (Isl1-derived) cells revealed a downregulation of SHF and CM markers including *Isl1, Tbx1, Mef2c, Fgf10, Hand2, Nkx2-5, Tbx20, Myl2, Cited2* in the presence of HG. At E11.5, *Tbx1, Mef2c, Hand2, Nkx2-5, Myl2,* and *Cited2* remained downregulated in matHG-exposed GFP⁺ cells; however, there was an upregulation of *Isl1, Fgf10* and *Tbx20* transcripts compared to CNTRL GFP⁺ cells. Therefore, reduced expression of genes encoding CM lineage-specifying TFs at E9.5 and upregulation of SHF progenitor markers at E11.5 in matHG-exposed embryos suggest a potential mechanism for CM differentiation defects contributing to an increased risk of CHD. Together, cell proliferation and gene expression studies in Isl1-derived GFP⁺ cells reveal that the SHF-progenitor cells are sensitive to the matHG environment which results in defective CM differentiation and contributes to reduced proliferation.

**Maternal HG perturbs *Isl1*-dependent gene regulatory network.** To build upon our findings from single-cell expression data and matHG-induced cell fate mapping, we assessed HG-mediated changes in the *Isl1*-dependent gene regulatory network (Isl1-GRN). We reconstructed Isl1-GRN with a priori knowledge of

protein–protein interactions (PPI) and known genetic association with supporting experimental evidence embedded in STRINGv11.5 database[62,63]. In conjunction with the DEGs in SHF (scRNA-seq data), we created a list of 34 genes (in mice and humans) including lineage specifying TFs, fibroblast growth factors (Fgfs), and epigenetic modifiers previously reported by Black, B (2007) and from the STRINGv11.5 database[30,62,63]. In vivo clonal analysis by Evans and coworkers have found a broader contribution of *Isl1-Cre*-expressing descendants to the heart[64]. The components of this GRN are known to function as part of an Isl1-dependent core network for RV and OFT development (Fig. 6a, b). The mouse and human PPI networks (enrichment $p$ value <1.0E-16) were created using STRINGv11.5 with Isl1 as a node (Fig. 6a, b). Functional enrichment analysis of this GRN revealed changes in biological processes associated with pattern specification, cell fate commitment and cardiac chamber morphogenesis in both species. Using the WikiPathways database[65], we showed Bmp signaling, pluripotency genes, neural crest differentiation, Id signaling and adipogenesis were mostly affected in murine Isl1-GRN. Human homologs of the genes showed CHD-associated genes (from DISEASES database[63]) and WikiPathways linked to mesodermal commitment, cardiac progenitor differentiation, BMP signaling, 22q11.2 copy number variation syndrome, and ventricular septal defects (Fig. 6c). Violin plots showed significant gene expression differences in Isl1-GRN components including *Isl1, Mef2c, Id2, Pdgfra, Pitx2, Mpped2, Foxp1, Crabp2, Tgfbi, Irx3, Bmpr2,* and *Hes1* between CNTRL and matHG exposed E9.5 SHF ($P_{adj} < 0.05$) (Fig. 6d).

To further evaluate the effect of matHG on Isl1-GRN expression in vivo, we quantified protein expression of Hand2, Nkx2-5, and Mef2c. Gene expression at the transcript level was quantified in CNTRL vs. matHG-exposed *Isl1-Cre⁺; Rosa^mTmG/+* E9.5 hearts (Fig. 5p). Co-immunostaining with Hand2 and GFP revealed significant downregulation of the protein expression in matHG-exposed E9.5 OFT (Fig. 7a–c, $p = 0.0005$). Similarly, Nkx2-5⁺GFP⁺ expression was significantly downregulated in the OFT and RV (Fig. 7d–f, $p = 0.012$), signifying the role of SHF progenitors in matHG-induced conotruncal defects. Recent scRNA-seq studies in *Hand2*-null embryos have shown the failure of OFT-CM specification, whereas RV CM were shown to be specified, but failed to properly differentiate and migrate[43]. Earlier studies have also demonstrated that *Nkx2-5* is required for SHF proliferation through suppression of Bmp2/Smad1 signaling and negatively regulates *Isl1* expression[66,67]. Based on scRNA-seq data, *Mef2c*-expression in C57BL6/J (*wt*) hearts was higher in matHG-exposed E9.5 SHF cluster compared to CNTRL hearts. Whereas qRT-PCR analysis in GFP⁺ cells and immunohistochemical staining in matHG-exposed E9.5 *wt*

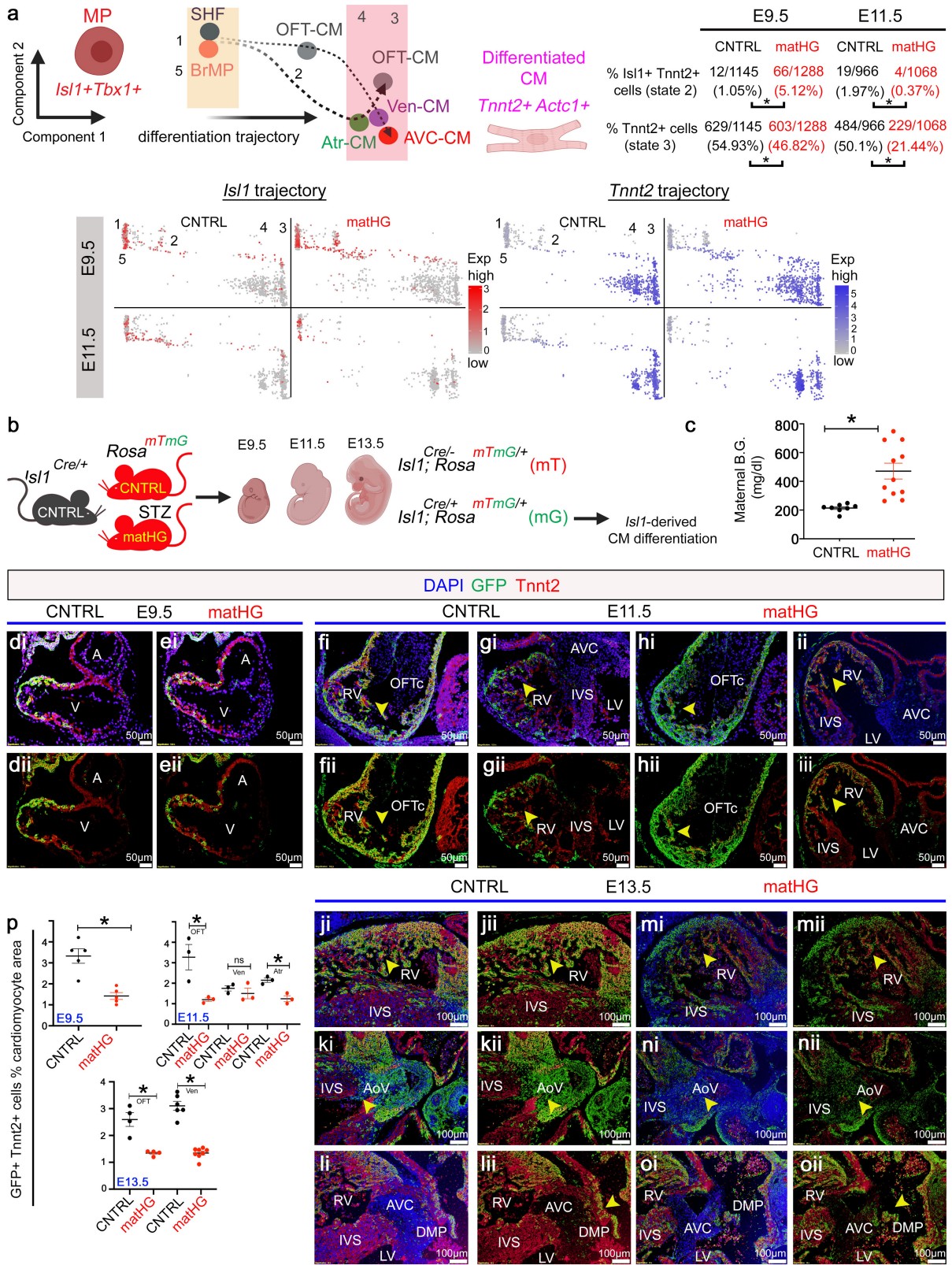

and *Is1l-Cre+; Rosa^mTmG/+* hearts have demonstrated that Mef2c protein expression was significantly downregulated in the distal OFT in response to matHG (Fig. 7g–k, *p* = 0.015 and 0.003). This may explain that this anterior SHF marker may denote only a subset of the Isl1 domain and represent a separate subdomain within the SHF and needs further investigation under HG setting. Finally, the quantitative evaluation of HG-sensitive Isl1-GRN and its link to CM

defects led to our proposed "matHG-induced CHD model" (Fig. 7l), where matHG perturbs Isl1-GRN in the SHF progenitors, leading to impaired CM differentiation and increases the risk of CHD.

## Discussion

In this study, we report two key findings based upon scRNA-seq data generated in matHG-exposed embryonic hearts. First, we

**Fig. 4 Exposure to matHG impedes SHF-derived cardiomyocyte differentiation. a** Schematics represent the trajectory of *Isl1+Tbx1+* multipotent SHF and *Tnnt2+Actc1+* CM. Feature plots denote the gene expression profile of *Isl1* (SHF-marker) and *Tnnt2* (CM-marker) in CNTRL vs. matHG-exposed E9.5 and E11.5 hearts across five pseudotime states (state 1–5) using Slingshot. Each dot represents single cells and color intensities (red for *Isl1* and blue for *Tnnt2*) display gene expression levels. The number and percentages of *Isl1+Tnnt2+* cells in state 2 (less differentiated CM) and *Tnnt2+* cells in State 3 (most differentiated CM) were quantified and compared between two groups using the chi-square test with Yate's correction. **b** Schematics represent the murine model of diabetes to study SHF-descendants in response to matHG. *Isl1-Cre+/−* males bred to CNTRL, and STZ-treated matHG *RosamT/mG* reporter females. Embryos were collected at E9.5, E11.5 and E13.5 stages to assess the effect of matHG on Isl1-derived CM differentiation. Cre− littermates are used as an internal control. **c** Statistical comparison of maternal B.G. levels in CNTRL (n = 9 litters; 217.7 ± 27.7 mg/dl) vs. STZ-treated *RosamT/mG* (n = 11 litters; 471.3 ± 182.9 mg/dl, * indicates p value <0.0001) dams at the time of embryo harvest. **d–o** Lineage-tracing in CNTRL and matHG-exposed *Isl1Cre+; RosaTmG/+* embryos compare Isl1-derived GFP+Tnnt2+ CM (yellow cells, shown in arrowheads) in E9.5 (di–ei), E11.5 (fi–ii) and E13.5 (ji–oi) hearts. (dii–oii) Co-immunofluorescence images demonstrate GFP+ (Isl1-derived, in green) and Tnnt2+ (CM, in red) cells in E9.5, E11.5 and E13.5 embryonic hearts exposed to CNTRL and matHG-environment. Nuclei stained with DAPI are shown in blue. **p** Quantification of *Isl1Cre+; RosaTmG/+* genetically labeled GFP+ cells co-stained with Tnnt2+ CM at three developmental stages (n ≥ 3 per timepoint/maternal diabetic condition) exposed to CNTRL and matHG environment. * Indicates p value = 0.001 in E9.5, p = 0.029 (OFT), and p = 0.007 (Atr) in E11.5, p = 0.003 (OFT) and p < 0.0001 (Ven) CMs in E13.5 CNTRL vs. matHG-exposed embryos. SHF second heart field, CM cardiomyocytes, CNTRL control, HG hyperglycemia, A atria, V ventricle, LV left ventricle, RV right ventricle, OFT outflow tract, IVS interventricular septum, AoV aortic valve, AVC atrioventricular cushion, DMP dorsal mesenchymal protrusion. Data presented as mean ± SEM; Statistical comparisons made between CNTRL and matHG groups by unpaired t-test with Welch's correction using GraphPad Prism 9. Scale bars: 50 µm (di–ii, dii–iii) and 100 µm (ji–oi and jii–oii). Schematic diagrams in (**a**, **b**) created with BioRender.com.

have segregated the transcriptional responses in matHG-exposed hearts at single-cell resolution. Second, we extended the computational reconstruction of CM differentiation trajectories to demonstrate that the matHG environment alters the gene regulatory network in *Isl1+* SHF cells and CM subpopulations at the early stages of heart development. Using in vivo cell-fate mapping, we were able to demonstrate that *Isl1+* SHF-progenitor cells are sensitive to matHG exposure, which leads to impaired CM differentiation and proliferation. The molecular interactions of Isl1-GRN obtained from single-cell expression data suggests its regulatory role in matPGDM-induced CHD in fetuses. In summary, we provide the experimental evidence of developmental toxicity of matHG on multiple cell lineages and focused our work on impaired CM function derived from the *Isl1+*SHF lineage.

Genetic alterations of lineage specifying TFs, growth factors and signaling molecules are considered to disrupt the spatio-temporal regulation of complex three-dimensional heart structures. However, interaction of multiple genetic and environmental factors is still considered as the primary etiology of ~85% of CHD[19,68]. Several of these CHD-causing variants are well studied in murine heart development, but the environmental etiology remains to be evaluated. Studies have shown that complete deletion of some causative genes results in embryonic lethality while heterozygous mice are often unaffected[69,70]. Recently, disease modeling of *GATA4* and *TBX5* pathogenic variants in human iPSC-derived CM and iPSCs-derived EC from patients with bicuspid aortic valve and calcific aortic valve disease with *NOTCH1* haploinsufficiency have predicted that CHD-linked GRN is sensitive to gene dosage[71–73]. These studies strongly suggest the importance of cell type and gene dosage in normal cardiac development. Like gene dosage, the impact of the matHG dosage is also of high clinical relevance. Variability in the adverse matHG environment in the presence or absence of genetic variants could similarly influence the disease penetrance and contribute to phenotypic severity observed in CHD patients. The clinical manifestations of matPGDM-induced CHD are variable, but the relative risk of CHD occurrence remains similar among the offspring of type 1 and type 2 diabetic mothers[16], suggesting HG as a potential teratogen. Infants are born with septal defects, conotruncal defects and heterotaxy at significantly higher incidence to mothers with PGDM[16,74]. However, these studies fail to differentiate between the contribution of genetic variants and the maternal environment linked to CHD. Studies using ex vivo and avian models have confirmed that matHG can cause cardiac defects[74,75]. In an experimental model of

matPGDM, we and others have established a gene-environment interaction and demonstrated an increased rate of matHG-induced CHD with genetic haploinsufficiency of cardiac genes[13,24,36]. Although the chemically (STZ) induced animal model of matPGDM strongly mimics human type 1 pathology, the effect of matHG could be applied universally to all pregestational diabetes diagnoses in future human-model studies. Additional mechanistic studies are required (i) to identify HG-sensitive cardiac regulatory genes and signaling pathways that may be potential therapeutic targets, (ii) to distinguish the role of HG-sensitive GRN in first and second myocardial cell lineages, and (iii) to model HG-sensitive genes related to abnormal cardiac development and CM dysfunction in the setting of matHG. The heterogeneity of cardiac lesions in the offspring of diabetic mothers also suggest a pleiotropic effect of matHG on diverse cell types. Prior in vivo studies with *wt* embryonic hearts led us to compare the breadth of cardiac phenotypes that occur under the direct influence of the matHG environment[24]. Notably, the preponderance of conotruncal CHD, ranging from failure of heart tube extension to arterial pole alignment defects, including DORV, and tetralogy of Fallot[29], suggest an SHF-origin. However, the toxicity of matHG on SHF lineage commitment remain unclear.

Single-cell multi-omics platforms have enabled gene profiling in humans and mice at an unprecedented resolution to reveal progenitor cell specification and their fate choice[76–79]. In this study, we uncovered transcriptional differences in CNTRL vs. matHG-exposed E9.5 and E11.5 hearts using 10XscRNA-sequencing. These two developmental timepoints were chosen to understand matHG-induced molecular changes after cardiac looping and segmentation to chambers. The differences in distribution of cell types between CNTRL and matHG-exposed mesodermal and neural crest cell populations corroborate the pleiotropic role of matHG exposure on cardiac development. Based on the manifestation of CHD in the offspring of the diabetic mothers, we found that a population of *Isl1+* cells that marks the SHF show significant gene expression changes with matHG. SHF gives rise to the OFT, RV, and portion of the atria, where the expression of *Isl1* is regulated as the cells adopt a differentiated phenotype[80]. In human studies, loss-of-function variants in *ISL1* alone or in synergy with *MEF2C*, *TBX20* or *GATA4* were shown to contribute to CHD and dilated cardiomyopathy[81]. Mutations in *ISL1* were found to be associated with maturity-onset diabetes of the young and type 2 diabetes patients[81,82]. In this study, we demonstrated that the *Isl1+* MP

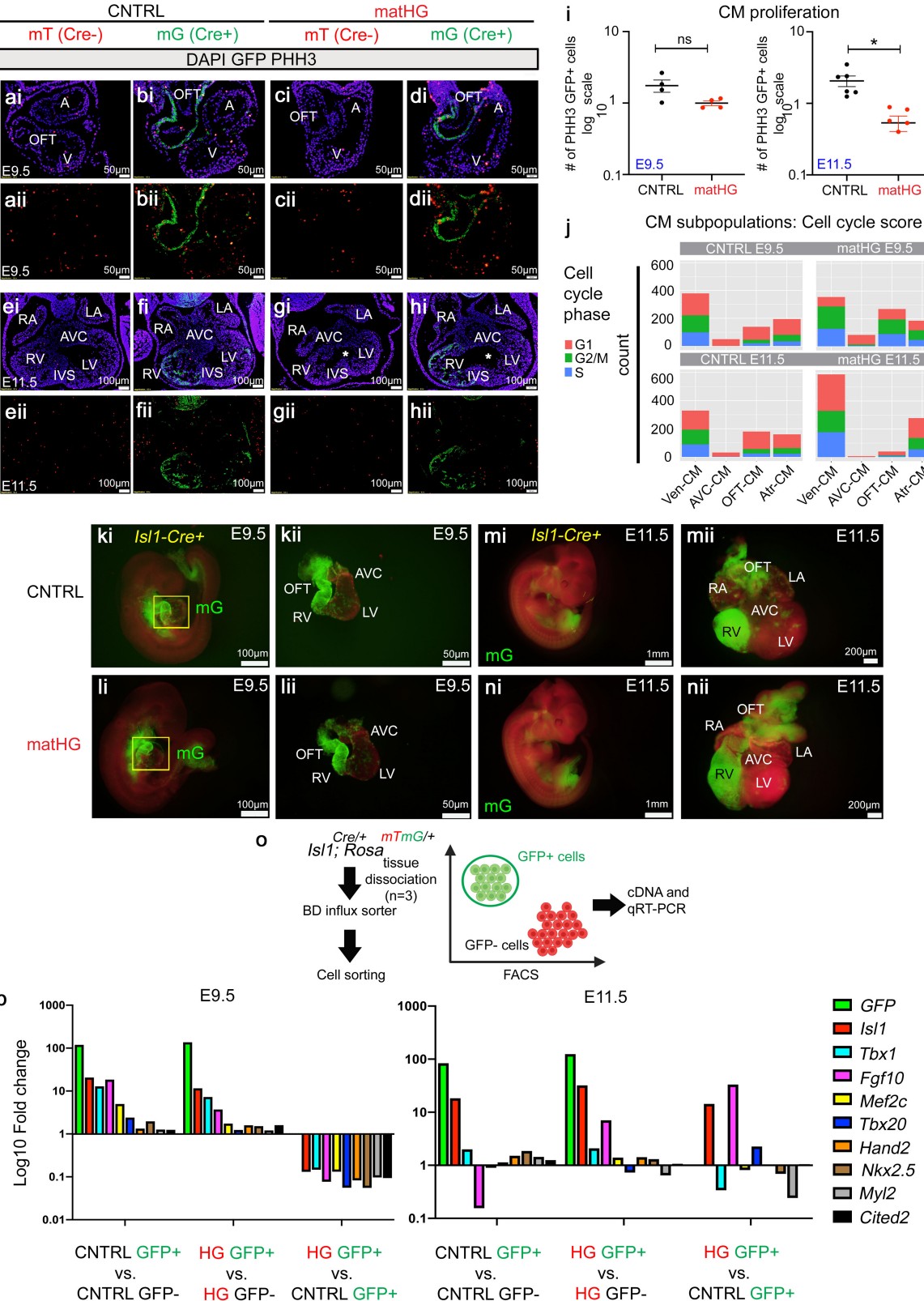

and *Tnnt2*⁺ CM populations subjected to intrauterine matHG have significant differences in gene expression compared to other cell types identified in scRNA-seq. The CM is the fundamental work unit of the heart. Although CM fate is a developmental end point, a diversity of CM subtypes exists within the heart[83]. After ballooning morphogenesis, the expansion of chamber-specific CM is followed by delamination and trabeculation, which

produce clear distinctions between the Atr-CM, Ven-CM, OFT-CM, and AVC-CM[42]. Therefore, we performed subclustering and pseudotemporal ordering of the MP-CM clusters to assess the effect of matHG on CM subpopulations. Our data identified gene expression differences in *Six1*⁺ *Fst*⁺ BrMP, *Isl1*⁺ *Tbx1*⁺*Osr1*⁺ SHF and *Bmp4*⁺*Rgs5*⁺ OFT, *Bmp2*⁺*Rspo3*⁺ AVC, *Myl2*⁺*Pln*⁺ Ven and *Nr2f1*⁺*Kcna5*⁺ Atr-CM subpopulations in matHG-

**Fig. 5 MatHG impairs cardiomyocyte proliferation and lineage specifying transcription factors. a–h** Immunofluorescent co-staining of PHH3 (red) and GFP (green) show Isl1-derived CM proliferation in CNTRL vs. matHG exposed E9.5 and E11.5 Cre$^-$ (mT) vs. Cre$^+$ (mG) cardiac tissues ($n \geq 4$ embryos per timepoint/maternal condition). The GFP$^+$PHH3$^+$ cells are quantified in (**i**), presented on a log$_{10}$ scale. Nuclei stained with DAPI are shown in blue. Data presented at mean ± SEM. Statistical comparisons made between CNTRL and matHG exposed E9.5 ($p$ value = 0.072) and E11.5 ($p$ value = 0.002) embryos by unpaired $t$-test using GraphPad Prism 9, ns non-significant and * indicates two-tailed $p$ value < 0.05. **j** Cell count and regression analysis on CM subtypes (OFT, AVC, Atr and Ven-CM) using Seurat reveal count in cell-cycle phases (G1, S, and G2/M) in E9.5 and E11.5 embryonic hearts exposed to CNTRL vs. matHG. **k–n** Representative fluorescence images show the endogenous GFP expression in CNTRL and matHG exposed E9.5 and E11.5 Isl1Cre$^+$; Rosa$^{mTmG/+}$ (mG) embryos and in microdissected hearts. **o** Schematics show whole hearts dissociated into single cells by enzymatic digestion. GFP$^+$ and GFP$^-$ cells were sorted using FACS and relative gene expression levels (Log$_{10}$ fold change) measured by SYBR green-based qRT-PCR analysis. **p** Comparisons between CNTRL GFP$^+$ vs. GFP$^-$ cells and matHG GFP$^+$ vs. GFP$^-$ cells show enrichment of GFP and other SHF/CM markers (*Isl1, Tbx1, Fgf10, Mef2c, Tbx20, Hand2, Nkx2.5, Myl2* and *Cited2*). Colors indicate gene names. Comparative gene expression analysis was performed in CNTRL GFP$^+$ vs. matHG GFP$^+$ cells (Isl1-derived cells) at the E9.5 and E11.5 stages. At least three independent embryos/timepoint/maternal diabetic status were used for qRT-PCR analysis. Data presented as Log$_{10}$ (fold change) normalized to endogenous Gapdh. Statistical comparisons between groups were made in GraphPad Prism 9. CNTRL control, HG hyperglycemia, A atria, V ventricle, LV left ventricle, RV right ventricle, AVC atrioventricular canal, IVS interventricular septum, OFT outflow tract, FACS Fluorescence-Activated Cell Sorting, qRT-PCR quantitative real-time polymerase chain reaction, SHF second heart field, CM cardiomyocytes. Scale bar: 50 μm (ai–di, aii–dii), 100 μm (ei–hi, eii–hii), 100 μm (ki, li), 50 μm (kii–lii), 1 mm (mi, ni), and 200 μm (mii, nii).

exposed hearts compared to CNTRL hearts. Notably, matHG-induced higher expression of Tbx1 in myogenic progenitors of branchial arches and a reduction in SHF cells suggest spatial dysregulation of these cardiac progenitor cells may contribute to the risk of CHD (Supplementary Fig. 14a–g). Monocle2 and Slingshot-based pseudotime trajectory analysis enabled us to identify differences in (i) progenitor-like States 1 and 5, (ii) intermediate and less differentiated OFT, and Ven-CM at State 2 and (iii) differentiated Atr-CM and Ven-CM at States 3 and 4. State-specific gene-set enrichment analyses have revealed alterations in PPAR, BMP, focal adhesion-PI3K-Akt-mTOR, adipogenesis, embryonic stem cell pluripotency pathways, calcium signaling, myometrial relaxation, and contraction pathways, HIF1α and p53 signaling, oxidative phosphorylation, pyruvate metabolism, and glycolysis/gluconeogenesis in response to matHG exposure. These metabolic changes found in E9.5-E11.5 CM might lead to delayed mitochondrial maturation under matHG. By comparing the mitochondrial morphology at E9.5 and E13.5 myocytes, studies by Hom et al; have shown that an open mitochondrial permeability transition pore is essential to drive the maturation of mitochondria and CM differentiation and that mitochondrial structure and function is dynamic during cardiac development[84]. A recent study by Solmonson et al; have applied in vivo isotope tracing and metabolomics approach to reveal compartment-specific differences in glucose-6-phosphate and other metabolites. Their data reveals the contribution of maternally derived nutrients to embryonic glucose metabolism and identified metabolic transition at E10.5-E11.5[85]. This method will inform our understanding of the developmental consequences of intrauterine metabolic defects on metabolic transition in the intact fetoplacental unit.

Reconstruction of the Isl1-GRN provides a list of putative candidate genes/pathways to evaluate gene-environment interaction in cardiac progenitor cells and CM differentiation to facilitate our understanding of matHG-induced CHD. Because matHG alters chromatin accessibility, and the Isl1/Ldb1 complex can orchestrate genome-wide chromatin organization to instruct the differentiation of multipotent cardiac progenitors, we theorize that matHG exposure impacts long-range chromatin interactions with *Isl1*-interacting loci and results in gene dysregulation[24,86,87]. Additionally, our genetic cell fate mapping studies imply that the SHF-derived CM are more HG-sensitive, and that CM differentiation is affected in the setting of matHG exposure. Here, we also demonstrated significantly reduced proliferation of Isl1$^+$-SHF-derived CM by E11.5 in matHG exposed embryos compared to CNTRL, validating prior studies that showed impairments in

CM proliferation at E13.5 in matHG exposed embryos[24,39]. Future BrdU pulse-chase experiments will test whether the proliferative capacity of CM precedes the delayed CM differentiation in reporter mouse lines to affect cardiac function. Several genes encoding structural proteins associated with both cardiomyopathies and CHD were also found to be dysregulated in matHG-exposed embryos. For example, reduction of Tpm1$^+$/Nkx2-5$^+$ protein expression was detected in matHG-exposed E11.5 hearts compared to CNTRL animals, also presented with trabecular disorganization and thin ventricular wall (Supplementary Fig. 15a–h).

In addition to transcriptional changes associated with MP-CM, the scRNA-seq data revealed differential gene expression patterns in endocardial/endothelial, mesenchymal, epicardial and neural crest cell lineages. For example, EC dysfunction in mothers with diabetes, obesity, and hypertension affect placental vascular supply and has been shown to harm fetal circulation and developmental pathways that can persist after birth[23,88–90]. In this study, we also performed DEG and functional enrichment analysis for EC, FM, EP, and NC clusters in CNTRL and matHG-exposed hearts (Supplementary Data 1–4). Our data suggested transcriptional changes in the genes associated with endothelial dysfunction such as in extracellular matrix (ECM) organization, angiogenesis, tube formation, OFT morphogenesis and heart valve development (Supplementary Fig. 16a–i and Supplementary Fig. 17a–g). The EC-FM genes including *Notch1, Klf4, TGFb, Snai2, Sox9, Vcan, Twist2, Pdlim3, Gata6, Bmp4, Msx1*, and *Pax3* were downregulated with matHG, and have been implicated in endothelial to mesenchymal transition (EMT), mesenchymal cell differentiation[22,24,39]. Significant reduction of FM cells and downregulation of EMT genes - *Vcan Sox9, Postn*, and Bmp/Tgfb signaling were detected in matHG-exposed E9.5 and E11.5 hearts (Supplementary Fig. 16e–i). Proepicardium-derived cells exhibit a migratory event at E9.5 in mice and are known to invade the myocardium and the endocardial cushions to generate a majority of vascular SMC and cardiac fibroblasts in the heart[91,92]. In the Tbx18$^+$ EP cluster, we found that the expression of *Wt1, Dcn*, and *Upk3b* was significantly downregulated with matHG and Wnt signaling inhibitor, and *Sfrp1* was upregulated at E9.5 (Supplementary Fig. 17e). Quijada P et al.; have demonstrated that EP-EMT is followed by an activation of Fn1 expression is necessary for ECM production and cell migration[92]. By immunohistochemical staining, we showed that matHG reduced Fn1 protein expression in E13.5 hearts (Supplementary Fig. 17f). Thus, EP maker-gene expression data implies potential migratory defects in proepicardial derived mesothelial cells in response to matHG.

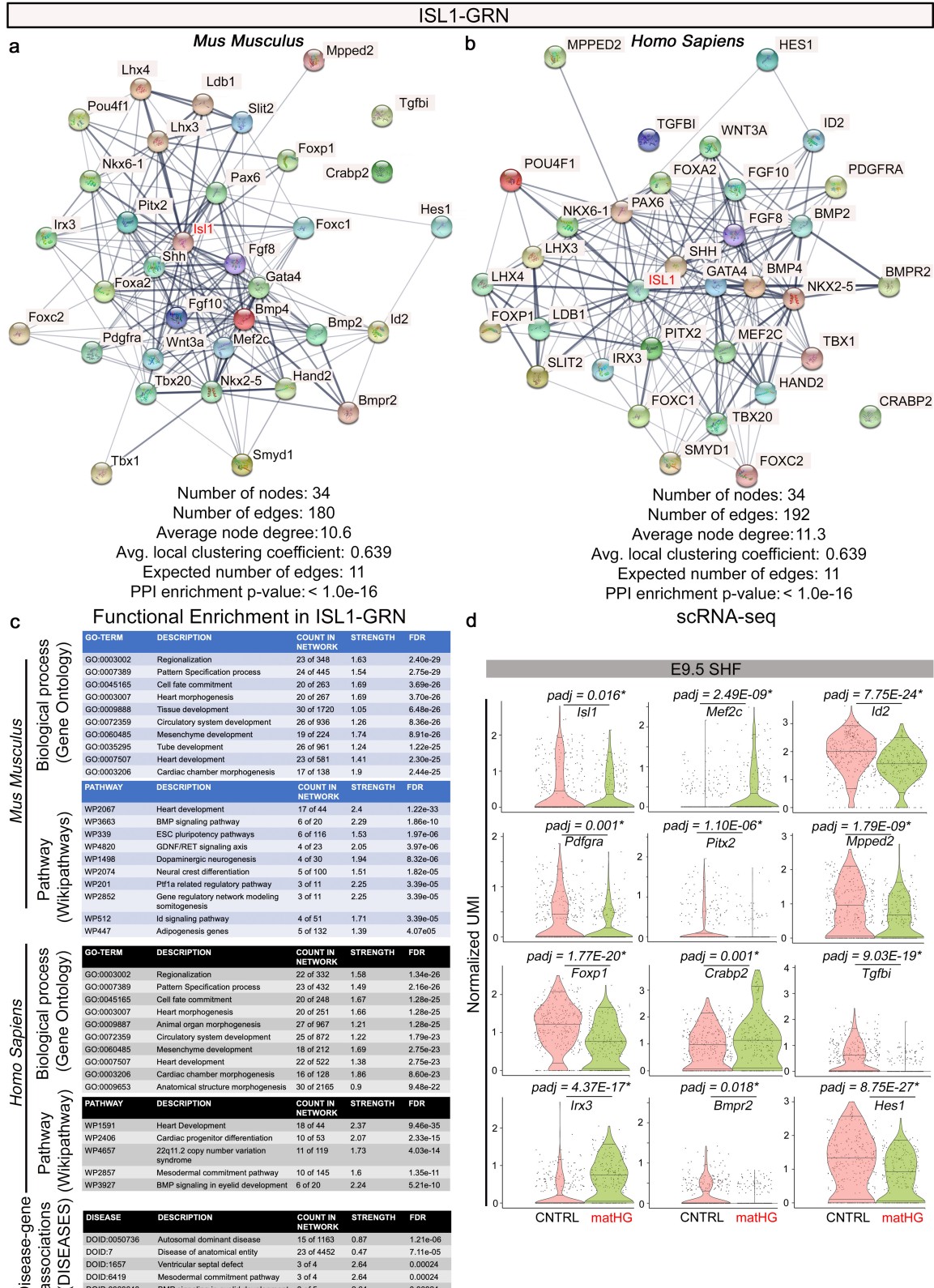

Apart from CHD, it is well established that matPGDM is significantly linked to neural tube defects in embryos and neuropsychological deficits in infants[93]. Here, we observed (i) significantly higher expression of *Dlx2*+, and *Dlx5*+ NC cells in matHG-exposed E9.5 hearts, which was diminished by E11.5 and (ii) downregulation of Pax3 expression in matHG-exposed E11.5 embryos compared to CNTRL (Supplementary Fig. 17e, g). A

recent scRNA-seq study by Soldatov R et al., in E8.5–E10.5 *Wnt1^Cre/R26R^Tomato* mouse embryos highlighted the branching trajectory of differentiating NC cells, where expression of *Dlx6* is followed by activation of *Msx2*, *Hand2*, and other cardiac markers including *Hand1* and *Gata6*[94]. Therefore, further cell fate-mapping studies in Wnt1 lineage are required to explain the overabundance of NC-derived mesenchymal population in

**Fig. 6 Perturbations in the Isl1-gene regulatory network are intrinsic to matHG-induced CHD. a, b** Identification of dysregulated genes in matHG-exposed SHF and reconstruction of Isl1-GRN with 34 proteins with known genetic interactions and experimental evidence using STRING database, and Black B, Semin Cell Dev Biol (2007)[24] mapped for *Mus musculus* and *Homo sapiens*. STRING output of the identified dysregulated TFs show predicted associations of Isl1-interacting proteins with protein–protein interaction (PPI) enrichment *p* value <1.0e-16. Network stats are highlighted, and line thickness indicates the strength of evidence data in the STRINGv11.5 database. Isl1 is highlighted in the PPI network (in red). **c** Species-specific enriched GO terms (biological processes), enriched pathways (WikiPathways database) and disease-associated genes (DISEASES database) are visualized. Count and strength of network associated with GO-terms, pathways, and disease-associated genes sorted as descending negative FDR adjusted *P* value of enrichment analysis. **d** Violin plots generated through Ryabhatta app show normalized UMI of Isl1-GRN components in CNTRL vs. matHG-exposed SHF at E9.5. This data (as of version 4.0 of Seurat) is log normalized expression from https://natian-and-ryabhatta.web.app/plotting-options.html#violin-plot and plots show median (center, horizontal line), 5th and 95th quantiles values. Show points on the Violin plot option display the cells that show expression present within the cluster. * Indicates DESeq2 $P_{adjusted}$ (padj) value ≤0.05. CNTRL control, matHG maternal hyperglycemia, SHF second heart field, PPI protein–protein interaction, GO gene ontology, GRN gene regulatory network, FDR false discovery rate, UMI Unique Molecular Identifier, scRNA-seq single-cell RNA sequencing.

matHG-exposed E9.5 hearts and to determine whether a common pathway leads to both conotruncal CHD and neural tube defects. The unidentified population of the cells (in cluster 11) was found to express primarily immune cell marker genes. However, there were additional markers suggested that this cluster might constitute doublets or multiplets. Analysis of this cell type, if determined not to be doublets, would be interesting from an immune response standpoint. However, its characterization is beyond the scope of this study.

While this study demonstrates matHG mediated changes in mouse cardiac transcriptomes at the single-cell level, we recognize that this study has several limitations. For single-cell transcriptomics, pooled samples were derived from embryos regardless of CHD status. In the early stages of development, there are not many structural defects that are visible without sectioning and staining the embryos to perform this stratification. Thus, some DEGs that were responsive to matHG may not have a direct role in causing CHD and might serve as a modifier. Conversely, the differential impact of matHG on CHD (cell type transition, DEGs) may have been under/overestimated with mixed samples. The second key limitation is that the data was obtained from a single litter for each sample. This limitation affects our ability to determine the effect of varying levels of matHG (multiple litters; HG dosage) on the development and severity of CHD. While single-cell transcriptional changes upon matHG exposure provided us a high-resolution view of the embryo's cellular response to this teratogen, they do not address the spatial transcriptional programs within the important anatomical context, such as in the OFT or chambers, nor directly address the potential contribution of cellular crosstalk leading to conotruncal and septal defects. Studies at earlier developmental time points and expanding on the number of CNTRL and matHG exposed hearts may prove essential in identifying the full spectrum of CHD. The transcriptional profiles of SHF progenitor cells and their effect on CM differentiation were further evaluated by cell-lineage-based studies. However, follow-up studies that examine the contribution of EP, EC, and NC-derived signals to drive CM differentiation upon matHG exposure will provide mechanistic insights into the environmental basis of CHD. Another limitation of the scRNA-seq analysis is processing power, as analyzing individual cells in a population requires hundreds of thousands to millions of single cells to be processed in a high throughput manner. While we used common markers to classify MP, CM, EC, FM, EP, and NC clusters, the ability to understand the complexity of a cell population requires careful analysis from all aspects. Therefore, the subpopulation of cells that we were unable to assign an identity would need further characterization. Future studies are required to carefully dissect the role of fetoplacental concentration gradient necessary for glucose transfer/utilization during the early stages of embryonic development. Better insights into

matHG and fetal B.G. levels will provide a robust quantitative measure in defining the abnormal cardiac outcome in the infants of diabetic mothers. Overall, our findings provide mechanistic insights into embryonic heart defects resulting from matHG, consistent with previous studies that revealed matPGDM-exposed offspring have a higher risk of conotruncal heart defects. It is vital to collate the cell-type-specific transcriptomic and epigenomic signatures in matHG-exposed embryos at multiple stages of cardiac development. These findings will elucidate how crosstalk between matHG and Isl1-GRN increase the risk of CHD in patients harboring identical genetic variants.

## Methods

**Generation of chemically induced maternal pre-gestational diabetes mellitus (matPGDM) murine model.** Wildtype (*wt*) C57BL/6 J mice were purchased from Jackson Laboratory (Stock Nos: 000664) for this study. A subset of six- to eight-week-old female mice (~15–18 g body weight) were used to chemically induce type 1-like matPGDM. Mice fasted for an hour before treatment and Streptozotocin (STZ, Fisher Scientific; NC0146241), dissolved in 0.01 mol/l citrate buffer (pH 4.5) was intraperitoneally injected at 75 mg/kg/bodyweight for 3 consecutive days, following a previously published protocol[24]. Seven days post-STZ treatment, mice fasted for ~8 h during the light cycle, and maternal B.G. levels were tested using the AlphaTrak veterinary blood glucometer calibrated specifically for rodents from tail vein blood (Abbott Laboratories). Blood glucose data were documented before initiating the timed breeding and during embryo collection. Mice with fasting blood glucose ≥200 mg/dl (11 mmol/l) were defined as HG status as previously published[24]. If the mice did not achieve the B.G. threshold after seven days, B.G. was re-tested after fourteen days of STZ injection. After confirming the HG status, mice were used for timed breeding. All the experimental mice were housed at the animal facility of Nationwide Children's Hospital (NCH), complied with all relevant ethical regulations for animal testing and research accordingly to Abigail Wexner Research Institute's Animal Resource Center policies, and the NIH's Guide for the Care and Use of Laboratory Animals (National Academies Press, 2011). All animal research has been reviewed and approved by an Institutional Animal Care and Use Committee (protocols: AR13-00056 and AR20-00029).

**Dissection of mouse embryos exposed to maternal control and hyperglycemic environment.** STZ-treated (matHG) and non-STZ-treated (CNTRL) females were timed bred overnight with *wt* C57BL/6 J males. Mice were maintained on a 12-hour-light/dark cycle, and in the morning, males and females were separated. When a vaginal plug was observed it indicated embryonic day (E)0.5. Pregnant mothers were monitored regularly and sacrificed for embryo harvest at E9.5 and E11.5 (for scRNA-seq analysis) and E9.5-E13.5 timepoints (for immunostaining purposes). The embryological staging and histological sectioning were followed as described by Savolainen, et al.[25].

**Tissue collection and workflow of single-cell RNA-sequencing.** To prepare scRNA-seq samples, the cardiogenic region was micro-dissected from multiple E9.5 and E11.5 embryos obtained from *wt* STZ-treated matHG and untreated CNTRL female mice. Maternal B.G. levels were tested before harvesting embryos to ensure diabetic status in CNTRL and STZ-treated females. The entire litter (n ≥ 6 embryos per timepoint per maternal condition) was used to prepare single cells. First, whole embryos were removed from the yolk sac, dissected in diethylpyrocarbonate-treated 1X ice-cold PBS, and placed in 1XPBS. Pooled hearts/timepoint/maternal condition were incubated in 1 mg/ml Collagenase II (Worthington; LS004176) and 1X TrypLE™ Select Enzyme (ThermoFisher Scientific, #12563029) for 15 min (for E9.5 embryos) and 25 min (for E11.5 embryos) at 37 °C dry bath with occasional stirring every 5 min for complete dissociation. The Collagenase/TrypLE solution

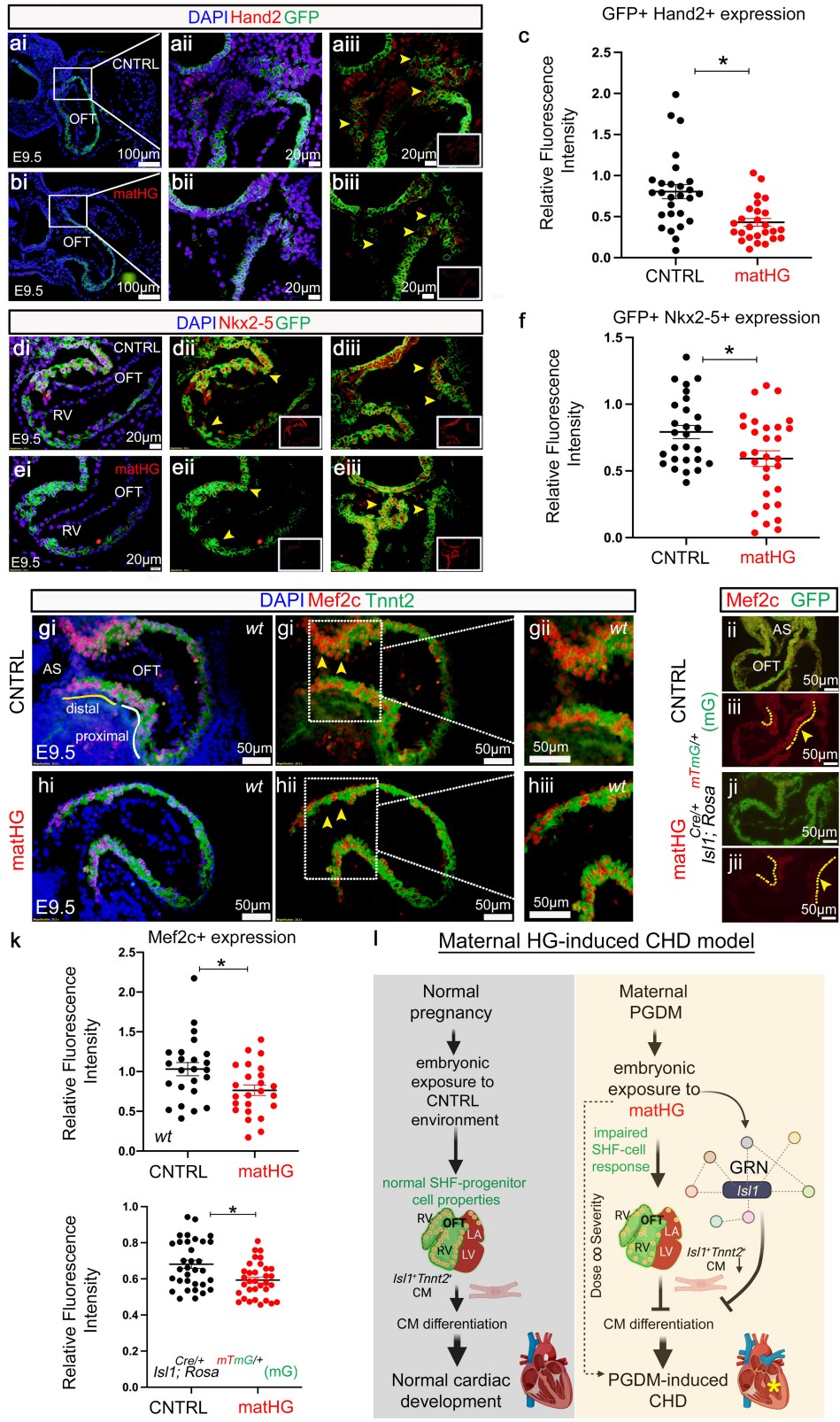

was quenched immediately with complete DMEM media supplemented with 10% FBS and pelleted down at 1000 rpm for 5 min at 4 °C. Cell pellets were dissolved in 0.04% bovine serum albumin, BSA (Fisher Scientific, #BP9703100) made in PBS and filtered through a 40μm cell strainer (BD Falcon, #352340), centrifuged at 1000 rpm for 5 min at 4 °C, and resuspended in 50 μl 0.04% BSA/PBS. Cell viabilities (>85-92%) were assessed using the Trypan blue (1450013; Bio-Rad) exclusion method on Countess™ II FL Automated Cell Counter (ThermoFisher;

AMQAF1000). Single-cell libraries targeting ~4000 cell recovery/sample were generated using 10X Genomics Chromium controller according to the manufacturer's instructions using Chromium Single Cell 3′ Reagent Kit (v2 chemistry; PN-120237). The cDNA and libraries were generated using the Chromium Single Cell 3′ Library & Gel Bead Kit v2 and Chromium i7 Multiplex Kit (10X, PN-120237, PN-120262) following the manufacturer's protocol. The quality and concentrations of cDNA from each sample were measured using High Sensitivity

**Fig. 7 Molecular dysregulation in SHF underlies matHG-mediated risk of CHD. a–f** Co-immunofluorescence staining shows Hand2 and Nkx2-5 (red, shown in insets) protein expression overlaying with GFP expression (green) driven by Isl1 in E9.5 hearts subjected to CNTRL and matHG conditions. The GFP+Hand2+ cells in E9.5 OFT in CNTRL and matHG-exposed embryos are shown in aii, bii (yellow arrowheads). GFP+Nkx2-5+ cells were shown in E9.5 OFT and RV-CM exposed to CNTRL and matHG environment (dii, diii, eii, eiii, yellow arrowheads). The relative fluorescence intensities of GFP+Hand2+ and GFP+Nkx2-5+ quantified in (**c**) (*p* value = 0.0005), and (**f**) (*p* value = 0.0121). Nuclei stained with DAPI are shown in blue. **g**, **h** Co-immunofluorescence staining of Mef2c (red) and Tnnt2 (green) protein expression in CNTRL vs. matHG-exposed E9.5 *wt* hearts. Yellow and white solid lines in (**g**) indicate distal and proximal OFT. White dotted lines in gii and hii indicates high magnification images shown in giii and hiii, respectively. Nuclei stained with DAPI are shown in blue. (**i**, **j**) Mef2c (red) and GFP (green) protein expression shown in E9.5 *Isl1Cre+; RosamTmG/+* (mG) hearts exposed to CNTRL and matHG environment. Yellow arrowheads and dotted lines indicate the downregulation of Mef2c in the OFT. The relative fluorescence intensities of Mef2c+ and GFP+Mef2c+ cells quantified in (**k**) (*p* value = 0.0151 and *p* value = 0.003, respectively). *n* ≥ 3 embryos/group in which multiple sections from each were used for quantification. Statistical comparisons made between CNTRL and matHG groups by unpaired *t*-test with Welch's correction using GraphPad Prism 9. Data presented as mean ± SEM and * Indicates two-tailed *p* value <0.05. **l** The proposed "matHG induced CHD model" shows molecular dysregulation in Isl1-GRN components leads to impairments in CM differentiation and increases the risk of CHD (schematics created with BioRender.com). A yellow asterisk indicates ventricular septal defects observed in matPGDM exposed embryos. CNTRL control, matHG maternal hyperglycemia, OFT outflow tract, RV right ventricle, AS aortic sac, CM cardiomyocytes, CHD congenital heart defect, GRN gene regulatory network, PGDM pregestational diabetes mellitus. Scale bars: 100 μm (ai, bi), 20 μm (aii–iii, bii–iii, di–iii, ei–iii), and (gi–ji, gii–jii and giii–jiii): 50 μm.

D5000 ScreenTape® on Agilent 2200 TapeStation. After adjusting four samples (CNTRL vs. matHG-exposed E9.5 and E11.5) to similar concentrations, single-cell cDNA samples were processed for library preparation and quantified using High Sensitivity D1000 ScreenTape®. Single-cell RNA-seq libraries were sequenced on the Illumina Hiseq4000 platform with 2 × 150 bp read length at the Steve and Cindy Rasmussen Institute for Genomic Medicine (IGM) in NCH. CNTRL and matHG-exposed E9.5 libraries were pooled and sequenced in the same lane to avoid batch effects. Similarly, CNTRL and matHG-exposed E11.5 single-cell libraries were pooled and sequenced in one lane to avoid batch effects. Sequencing parameters were selected according to the Chromium Single Cell v2 specifications. All libraries were sequenced to a mean read depth of at least 50,000 reads per cell.

**Single-cell RNA-seq data processing, quality control and unsupervised clustering**. We used the Cell Ranger 'mkfastq' function to demultiplex and convert Illumina '.bcl' output into fastq files. We mapped the fastq reads to the mouse genome mm10 (GRCm38.p6) and gene annotation downloaded from 10X Genomics (https://cf.10xgenomics.com/supp/cell-exp/refdata-gex-GRCh38-2020-A.tar.gz) using the Cell Ranger 'count' function and generated the gene-count matrix output. The Cell Ranger output was used to create a Seurat object using the R Shiny app Natian (available through www.singlecelltranscriptomics.org). The data processing pipeline and filtering cutoffs are described in Supplementary Fig. 1. Seurat object creation and processing steps described below were performed using Natian[95]. Natian is graphical user interphase used to run the command line functions available through the R-package Seurat. We created Seurat objects for each sample and added 'DevStage' and 'Maternal' meta information to each object using 'Add Timepoint/Treatment' button in Natian to define the developmental stage and the diabetes state of the dam, respectively. Then, the low quality cells or those representing doublets were excluded from our analyses using the following cutoffs: Number of genes expressed (*nFeature_RNA*) set to 500-7000 and percentage of mitochondrial gene expression relative to the total expression of the cell (*percent.mt*) ≤10%, we obtained 2823, 3239, 3836 and 3229 cells from CNTRL-E9.5, CNTRL-E11.5, matHG-E9.5, and matHG-E11.5 respectively. Gene expression was normalized, scaled and 50 principal components (PCs or dims) were calculated from the expression of the highly variable genes using Natian. Using a heuristic approach based on the elbow plot between the standard deviation calculated from each dim, we chose 20-25 dims for each sample to perform dimensionality reduction based on the Uniform Manifold Approximation and Projection (UMAP)[96] algorithm using Natian. Cells were clustered using the Louvain algorithm using a resolution determined based on the "clustree" diagram generated through Natian. Next, we combined the data using regularized negative binomial regression ('SCTransform' function in Seurat) and canonical correlation analysis (CCA) based integration in Natian. We also tried integration without 'SCTransform' function to examine the over-correction of cell types. There was no significant difference in the clustering of cells based on the two approaches. We performed dimensionality reduction and clustering of cells on the integrated data using a very broad clustering parameter (resolution = 0.2).

**Cell type identification, sub-clustering analysis and comparison to de Soysa et al., *wt* embryonic scRNA-seq data**. For scRNA-seq data, marker genes were identified for individual clusters using a minimum percent expression of 50% and log₂fold change threshold of 0.25 (log₂FC.threshold). From each cluster, the top 5 markers were selected based on average log₂FC and used to classify each cluster in Ryabhatta (available at www.singlecelltranscriptomics.org). The clusters were also cross-referenced with known cell-type-specific marker gene expression using publicly available wildtype scRNA-seq datasets at other embryonic time points[42,43]. All sub-clustering analyses (on multipotent progenitor cell and CM subpopulations) were processed using Ryabhatta using a similar number of principal components and resolution parameters.

To identify cell types and sub-cell-types populating individual clusters we used previously published embryonic heart single-cell data from E9.25 *wt* mice (GSE126128). The count matrix file GSE126128_AllTimePoints_WT_10X.csv.gz was downloaded from Gene Expression Omnibus. The meta data information containing cluster information, and time point information was downloaded from UCSC Cell Browser (https://cells.ucsc.edu/?ds=mouse-cardiac). The two datasets were used to generate a Seurat object using Natian. In Natian, the 'load the gene count matrix' button was used to start the process of creating the Seurat object. The data was loaded with the name GSE126128 and used to create a Seurat object. Then, the 'Add time point/treatment information' button was used to select the option to upload data using 'Load meta data from file'. The meta data information file from UCSC Cell Browser was then used to populate the meta data slot of the Seurat object. Next, quality control is performed using 'Perform initial QC analysis'. No cells are filtered as the data is already processed to remove outliers. Following this, a standard analysis of single-cell data using normalization, scaling data, and principal component analysis is performed using the 'Filter cells and perform PCA' button. We then used 20 dims (principal components) and a resolution of 0.5 to perform dimensionality reductions using UMAP and tSNE methods and cluster cells using the default Louvain algorithm using the 'Use PCs and cluster cells' button. To add sub-cluster data for the multipotent and myocardium populations, we used corresponding meta data files from UCSC Cell Browser. These two files were combined and used to update the 'meta.data' slot of the Seurat object using the 'Resume processing Seurat file', and 'Add time point/treatment information' button. This file was saved using the 'Save Curr File' button as an '.RDS' file.

This '.RDS' file was opened using Ryabhatta and split into 3 timepoints in the 'meta.data' slot using 'Edit Meta data' > 'Create subset objects'. The E9.25 Seurat subset object was integrated with the data generated by us using Natian. For the integration, we used 'Resume processing Seurat file' button to load the individual Seurat '.RDS' files. Then we used the 'Integrate Seurat files' button to select the E9.25 Seurat object from de Soysa et al. (GSE126128) and our data labeled Manivannan et al., 2022 (GSE193746). With 'Perform SC transformation on individual data' in the app selected, we integrated these two objects. The app performed a default PCA and dimensionality reduction analysis to yield an Integrated Seurat object. This Integrated Seurat object was then imported into Ryabhatta to generate Supplementary Fig. 2.

**Prospective and retrospective analysis of the number of cells required to capture rare cell types**. We used SCOPIT (https://alexdavisscs.shinyapps.io/scs_power_multinomial/) for power analysis to evaluate the number of cells needed to be captured in each sample[44]. For the prospective analysis, we changed the number of cells required parameter while keeping the probability of capture and frequency of the rarest cell type at 1% in 1 cluster. We then plotted the data as Cells per individuum (per individual sample) against the minimum number of cells per target cell type per individuum (per individual sample). We performed this analysis with 99% probability and 95% probability of capturing the set number of cells. Based on this, we calculated the expected number of a rare cluster (that occurs at 1% of the population) that can be captured at 95–99% probability.

For retrospective analysis, we computed the percentage of each cell type post marker identification and clustering analysis. In this case, we used the population frequency of cell types observed in control E9.5 as our observed frequency of each cluster. We calculated the probability of capturing at least 20 cells of the lowest frequency cluster (immune cells). Having captured 90 immune cells by sequencing 2741 cells after removal of outliers showed that the number of cells captured was sufficient to capture rare cell types with an observed frequency of ~1%.

**Genotype-free demultiplexing the scRNA-seq data using "souporcell"**. To separate pooled embryonic hearts *post hoc* sequencing, we used the genotype-free demultiplexing tool 'souporcell'[45]. This tool uses single nucleotide polymorphisms

(SNPs) identified from the single-cell transcriptome, the ambient RNA signal, and the number of sub-samples (clusters) pooled together to demultiplex samples. Souporcell uses the assumption that allele counts are derived using a binomial distribution of the alternate alleles shifted by the ambient RNA signal. The model is used to solve for the maximum-likelihood soup fraction with gradient descent. We used the souporcell singularity image from shub://wheaton5/souporcell and use the souporcell_pipeline.py script with the expected number of clusters to 8 for each sample, the --max_loci at 6144 (three times the default) to infer individual samples.

Each sample (CNTRL E9.5, CNTRL E11.5, matHG E9.5, matHG E11.5) was run independently through the souporcell process. The sample assignments from souporcell for each genotype (0–7) were then added to the corresponding Seurat object using Natian (Add Timepoint/Treatment button). We note that the genotypes for individual samples are different, and the same numbers (genotype 0–7) used to represent a genotype in two different samples do not indicate the same genotype.

**Differential gene expression, Gene Ontology enrichment, and protein–protein interaction analysis.** Cluster-specific differential gene expression analysis was performed between embryonic developmental stage and maternal diabetic status. Within each group, we combined counts obtained for each gene to produce three in silico replicates of the gene vs. expression count matrix. These in silico replicates were then analyzed using DESeq2 to identify differentially expressed genes between various conditions[97]. The statistical test for DESeq2 (pseudobulk) is the Wald test, with the *P* value adjusted for multiple testing using the Benjamini and Hochberg method. Owing to the high drop-out rate observed in the 10X drop-seq method, our approach combining counts reduces noise and brings the data closer to bulk-RNA seq data which are conventionally used in the DESeq2 pipeline[97,98]. The GO annotation of the up/down-regulated DEGs was performed using ShinyGO v0.741 software, a web-based graphical gene-set enrichment tool[99]. The biological process, affected by matHG exposure was represented as fold enrichment, the number of genes in each GO-term with -log10(FDR) values <0.05. Protein–protein Interaction (PPI) networks (mouse and humans) of identified Isl1-interacting proteins ($n = 34$ proteins) were created using STRINGv11.5. STRING database (https://string-db.org) is a curated knowledge database of known and predicted protein–protein interactions[62]. Most of the Isl1-GRN proteins were retrieved from published literature[30] and demonstrated an established link with each other in the interaction network.

**Pseudo-time trajectory analysis.** Cell trajectory analyses were performed on matHG-exposed E9.5 and E11.5 vs. CNTRL E9.5 and E11.5 MP and CM sub-populations using the Monocle 2 (http://cole-trapnell-lab.github.io/monocle-release/) in Ryabhatta. The pseudotime data was further used to generate smooth trajectory curves using Slingshot version 1.8.0 packages[56,57]. Differentially expressed genes were determined using the *FindAllMarkers* function in the Seurat package implemented in Ryabhatta for temporal ordering of these cardiac cells along the differentiation trajectory in response to CNTRL vs. matHG environment.

Briefly, to evaluate the temporal changes to the cell type observed using Natian processing and marker-based annotation, we extended dimensionality reduction analysis by drawing from Pseudotime analysis using monocle in Ryabhatta and UMAP-reduction performed in Natian using Seurat. We used Ryabhatta (running monocle2 functions) to perform pseudotime analysis, which involves a dimensionality reduction step using the DDRTree algorithm. The dimensions of this reduction are called "Component_1" and "Component_2" (analogous to UMAP_1 and UMAP_2). In this case, Component 1 closely tracks "pseudotime" calculated by the "orderCells" function in monocle2[54]. Therefore, we created a new dimensionality reduction using the calculated "pseudotime" as the actual "Component_1" and used the UMAP_1 dimension from the RunUMAP function (Seurat) reduction as "Component_2". The goal of this approach is to identify the relative changes of cell types tracked using UMAP_1 with the Component_1 (pseudotime) from monocle2[54]. To estimate the trajectory of the cells along with this dimensionality reduction, we used Slingshot. This analysis permitted tracking the different sub-clusters identified in the MP and CM clusters.

**Second heart field cell-lineage tracing in a murine model of maternal hyperglycemia.** For *Isl-1*-derived cardiac cell-lineage tracing studies, we purchased *Isl-1Cre*[+/−] male mice and double fluorescent *Rosa*[mT/mG] reporter mice from Jackson Laboratory (Stock Nos: 024242 and 007676). Diabetes was chemically induced in females as described earlier and maternal B.G. was tested fourteen days after STZ treatment. Mice with fasting B.G. ≥ 250 mg/dl were defined as HG in *Rosa*[mT/mG] strain. Adult males were bred with 6–8 weeks old STZ-treated and CNTRL homozygous *Rosa*[mT/mG] female mice. Following timed breeding, CNTRL and matHG-exposed *Isl-1Cre*[+]; *Rosa*[mTmG/+] embryos at E9.5, E11.5, and E13.5 were harvested for histologic analysis. *Cre*[-] littermates were used as internal controls. TdTomato and GFP fluorescence intensities of whole embryos were captured using an Olympus BX51 microscope and matched with Cre genotyping profile. Embryos were fixed in 4% paraformaldehyde for 24 h and changed to PBS the next day and sent to Morphology Core at NCH for embedding and transverse (7 μm) sectioning.

**Immunofluorescence and immunohistochemical staining.** For immunofluorescence (IF) staining, E9.5, E11.5, and E13.5 paraffin-embedded cardiac

sections were deparaffinized using xylene and grades of ethanol, followed by antigen retrieval using citrate-based Antigen Unmasking solution (H-3300, Vector laboratories) using standard protocols[24]. After permeabilization and blocking with 1% BSA in PBS-Triton X-100 for 1 hour, tissue sections were probed overnight at 4 °C with primary antibodies including rabbit and mouse α-GFP (1:250; Abcam, ab290 and 1:100; sc-9996), mouse α-Cardiac Troponin T (1:250; Abcam, ab8295), rat α-Endomucin (1:250; Millipore, MAB2624), rabbit α-Sox9 (1:250, Abcam, ab185230), rabbit α-Periostin (1:250; Abcam, ab14041), rabbit α-Transgelin or SM22-α (1:250; Abcam, ab14106), rabbit α-Fibronectin (1:250; Abcam, ab2413), rabbit α-Hand2 (1:250, Abcam, ab200040), rabbit and goat α-Nkx2-5 (1:250; CST-E1Y8H, 8792 and 1:100; sc-376565), rabbit α-Mef2c (1:250, CST, D80C1), mouse α-Tropomyosin (1:50; Clone CH1 DSHB) and mitosis marker, rabbit α-phospho-Histone H3 (PHH3; 1:250, EMD Millipore, 06-570). Following a series of washing, the sections were incubated with a donkey α-rat, α-rabbit, and α-mouse secondary antibodies conjugated to Alexa Fluor 594/488 for 1 hour at room temperature in the dark. After washing, sections were counterstained with Vectashield HardSet Antifade Mounting Medium with DAPI (Vector laboratories). The images were visualized using an Olympus BX51 fluorescence microscope.

For immunohistochemical (IHC) staining, embryonic heart sections were deparaffinized in xylene and rehydrated in grades of ethanol and 1 × PBS. Briefly, after antigen retrieval tissue, sections were incubated with 3% $H_2O_2$ diluted in water at room temperature for 10 min to quench endogenous peroxidase activity and blocked by 5% normal goat serum in 1 × TBS containing 0.1% Tween-20 (Sigma-Aldrich) for 1 h at room temperature to avoid nonspecific binding per manufacturer's protocol. Following this, sections were incubated with rabbit α-Tbx1 (1:200; Thermo Scientific, 34-9800) diluted in SignalStain Antibody Diluent (8112, Cell Signaling Technology) overnight at 4 °C. Normal IgG served as a negative control. Next, sections were washed with 1 × TBST and incubated with SignalStain Boost IHC Detection Reagent (HRP, rabbit, 8114, Cell Signaling Technology) for 30 min at room temperature. Sections were washed three times with 1 × TBST and visualized using the SignalStain DAB Substrate Kit (8059, Cell Signaling Technology) and imaged using Zeiss AxioImagerA2. All staining experiments were performed at least in three independent embryos from CNTRL and matHG exposure groups.

**Tissue dissociation and GFP+ cell sorting from Isl1-*Rosa*[mT/mG] embryonic hearts.** The embryos were harvested at the E9.5 and E11.5 stages. CNTRL and matHG-exposed hearts were isolated from mice carrying both tdTomato and GFP fluorescence signals. Isolated hearts were digested with collagenase and 0.25% trypsin-EDTA into single cells, and GFP+ and GFP− cells were sorted using FACS (FACSAria™, Becton Dickinson) at the NCH flow cytometry core. Sorted cells were collected into TRIzol (15596018, ThermoFisher Scientific) and stored at −20 °C until RNA extraction. Sorting experiments were performed on at least three pooled embryos per timepoint per maternal condition. At least three sets of pooled embryos for all conditions were used for qRT-PCR assay.

**RNA purification and quantitative real-time PCR.** RNA was extracted from flow-sorted GFP+ and GFP− cell populations exposed to CNTRL and matHG environment using TRIzol Reagent followed by chloroform-isopropanol extraction and purification as described earlier[24]. Then RNA was quantified spectro-photometrically and 500ng-1μg of total RNA was used for reverse transcription using the SuperScript VILO cDNA Synthesis Kit (11754-050, ThermoFisher Scientific). SYBR Green-based qRT-PCR was performed for SHF (*Isl1, Nkx2-5, Tbx1, Tbx20, Mef2c, Fgf10, Hand2*), and CM (*Myl2, Cited2*) markers using StepOnePlus™ Real-Time PCR System (Applied Biosystems). Mean relative gene expression was calculated after normalizing Ct values to *Gapdh* using the ΔΔCt method and presented as Log10(Fold change). Three independent replicates were performed after pooling three embryos per timepoint per maternal condition. Oligonucleotide sequences of these genes are provided in Supplementary Table 2.

**Statistics and reproducibility.** All the experiments were performed at least in triplicate and data are presented as the mean ± standard error of the mean (SEM) (see figure legends for additional information). Except for scRNA-seq data, statistical significance between CNTRL and matHG-exposed groups was determined by two-tailed unpaired *t*-test with Welch's post-hoc correction, Fisher Exact test, and Chi-square test with Yate's correction using GraphPad Prism 9 software package and https://www.graphpad.com/quickcalcs/contingency1/. Comparison between two groups were considered statistically significant when the two-tailed *p* value was ≤0.05. Image J (NIH) was used to quantify the IF and IHC images and all the plots were generated by GraphPad Prism 9.

**Reporting summary.** Further information on research design is available in the Nature Research Reporting Summary linked to this article.

## Data availability

The single-cell RNA sequencing data underlying this study have been deposited in NCBI Gene Expression Omnibus (GEO) database (GSE193746). The genotype data (vcf files) has been submitted to figshare repository. Th source data underlying Figs. 1e, 2c, 3b, d,

4c, p, 5i-j, p, 7c, f, k have been provided in Supplementary Data 10-21. Figure 6d has been plotted from E9.5 scRNA-seq data, using the Shiny apps https://natian-and-ryabhatta.web.app/plotting-options.html#violin-plot", provided in the NCBI GEO accession (GSE193746). Additional materials are available from the corresponding author upon request.

## Code availability

All the analyses were performed using standard protocols with previously described R packages[44,45,56,57]. The R Shiny apps, Natian and Ryabhatta, developed to process, analyze, and visualize our single-cell transcriptomic datasets are publicly available at www.singlecelltranscriptomics.org [40]. We have also compiled a free (open access) manual for all users found in https://doi.org/10.5281/zenodo.6914947. The R scripts are available upon request.

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

## Acknowledgements

The authors thank members of the Biomorphology core for histology support and The Steve and Cindy Rasmussen Institute for Genomic Medicine at Nationwide Children's Hospital for sequencing and preliminary bioinformatic analysis. Schematic diagrams were created with BioRender.com. The authors are grateful to Dr. Kedryn K. Baskin at The Ohio State University and Dr. Dennis Lewandowski in the Heart Center Editorial Service Core for helpful comments and reviewing the manuscript. This work was supported by funding from the American Heart Association and Children's Heart Foundation Career Development Award Grant 18CDA34110330 and Diabetes Research Connection (M.B.), funding from the National Institutes of Health/National Heart Lung and Blood Institute award R01HL144009 (V.G.) and a Postdoctoral Fellowship award T32HL098039 (S.M.).

## Author contributions

M.B. and V.G. conceived and designed the project and M.B. interpreted the experimental results with input from S.M. for scRNA-seq data. M.B. supervised the experiments with input from V.G. M.B. performed immunohistochemical characterization, generated single-cell libraries, acquired all the data and analyzed the experiments. S.M. performed bioinformatics analysis of scRNA-seq data, pseudotime trajectory analysis using Monocle and Slingshot and developed the Seurat-based graphical user interface, Natian and Ryabhatta. C.M. performed the murine studies including S.T.Z. injections, embryo collection, genotyping, sectioning and histological imaging, qRT-PCR analysis, and quantification with supervision by M.B. X.Z. performed initial scRNA-seq data processing. M.K. generated control E11.5 single-cell library. U.M. assisted in the analysis of histological images. M.B. wrote the manuscript with input from S.M., C.M., and all the other authors.

## Competing interests

Authors declare no competing interests. MB is an Editorial Board Member for *Communications Biology*, but was not involved in the editorial review of, nor the decision to publish this article.
