## [Peer Review File · Communications Biology]

Reviewers' comments:

Reviewer #1 (Remarks to the Author):

The authors obtained single-cell RNA-seq expression data of the hearts from maternal hyperglycemia (mathHG) model mice, and show that mathHG affects transcriptional dysregulation during heart development and especially disrupts the populations of Isl1+ second heart progenitors and Tnnt2+ cardiomyocytes. Although these findings are interesting, almost all findings in the manuscript are lead by single-cell RNA-seq data, which is derived from one mouse for each condition (CNTRL E9.5, CNTRL 11.5, mathHG E9.5, mathHG E11.5). First of all, considering the biological variability, the authors should obtain single-cell data from several organisms for each condition (biological replicates), and assess the biological variability of cell fractions and gene expression levels.

Other points.

1. In Figure 2C-E, the authors conducted GO analysis of DEG. I would like the authors to show the differences in expression levels of representative genes for example using violin plot or heatmap.
2. From line 264, the authors describe that given the contribution of MP subpopulations during heart development, these cellular differences suggest a potential role of glucose sensitivity in cells from the SHF and increased risk of CHD in the diabetic offspring. Please cite the relevant papers.
3. In Figures 2 and 3, the authors detected the differences in gene expression levels of multipotent progenitors (MP) and cardiomyocytes (CM) between CNTRL and mathHG. The authors should validate these transcriptional changes for example using RNA in situ hybridization.
4. In Figure 4D, the authors used UMAP to reduce dimensions for each cell, but I cannot understand how the authors calculate the component2 1 and 2 in Figure 4E. This seems strange.
5. In line 337, Fig. 5G should be replaced with Fig. 4G.
6. From line 352, the authors describe "we found higher number of Isl1+ Tnnt2 + cells at the less differentiated intermediate State 2 in mathHG-exposed E9.5, while the reduction in Tnnt2 + cells was observed in more differentiated CMs at E11.5 State 4 (Fig. 5A)." Please show this quantitatively.
7. How does mathHG affect the ISL1-associated gene regulatory network? Epigenetic regulation? Please cite some relevant article associated with possible underlying mechanisms in the Discussion section.
8. In line 365, Fig. 8A should be replaced with Fig. S8A.

Reviewer #2 (Remarks to the Author):

This is an interesting study using some state-of-the-art technologies. Manivannan et al. reported the embryonic response to mathHG environment, especially Isl1+ second heart field progenitors and Tnnt2+ cardiomyocytes. Then through Isl1-Cre+/- and RosamT/mG dual reporter mice, the authors confirmed that mathHG exposure leads to impaired cardiomyocyte differentiation. It has a well-thought logical structure and the figures and tables overall are reluctantly supporting the premise of the study.

However, there are some major concerns that arise from looking at the experimental data that need unambiguous clarifications or thorough revision before possible publication.

Major concerns:

1. There was only 1 sample at each group, which makes it very difficult to evaluate the differences among samples. This also probably led to accidental error. Therefore, the authors should increase sample size for single cell RNA sequencing if possible.
2. I addressed all of my concerns regarding data processing. Nearly half of the single cell RNA

sequencing data was discarded (Line 169-171), which could result in pronounced changes to cell constitution and accuracy of follow-up data analysis. Thus, the authors should detailly explain the motivations behind the data processing, and they also should include analyses to show that alternative processing of the data give rise to the same conclusions.

3. In this study, the authors keep only the cells of mesodermal and mesectodermal origin in further analysis. However, there remain endocardial/endothelial (EC) and epicardial (EP) in cluster definition, how to explain it?

4. As for the comparisons of the cellular proportions, rejecting large amounts of data and insufficient sample sizes make a low accuracy. Those results should be confirmed by other experiments.

5. Line 257-259, the CMs further divided into outflow tract (OFT), atrioventricular canal (AVC), atrial (Atr) and ventricular (Ven)-CM subpopulations. What basis is there for this?

Minor concerns:

1. The authors should add relevant references about CHD in line 63-65.

2. Line 74 "increased occurrence of CHD", consider revising to "increased occurrence of CHD in fetuses"

3. Line 81~82. I wonder if "transposition of great arteries, outflow tract (OFT) defects with normally related great arteries, double outlet right ventricle (DORV)" could be simplified as "conotruncal defects"

4. Please delete the "and their derivatives" in Line 100.

5. "*****and when deleted in mice demonstrate embryonic lethality and higher incidences of CHD", please check and correct it

6. In terms of language and style, the manuscript itself needs to make improvements. There are too many individual comments to list them all here.

7. I suggest that authors include a paragraph listing the limitation of this study including small sample size

8. In part "Maternal hyperglycemia triggers key molecular changes in cardiac progenitors and cardiomyocytes", what is the the experimental group and control group in each differential expression analysis should be clearly described to avoid any ambiguity.

9. Line 270-272 " Atr and Ven-CMs significantly higher in the E11.5 hearts", while Line 279-281 "the expression of Gja1+ Ven-CMs, Bmp4+, Rgs5+ and Dlk1+ OFT-CMs, Cacna2d2+, Tbx3+ AVC-CMs and Kcan5+, Nr2f1+, Nr2f2+ Atr-CMs were reduced with mathHG". Please check this inconsistency.

10. Figure 4D is quite confused for me. From this plot, each subpopulation seems to be in a unique state, this does not agree with Figure 4B.

11. The Figure 8A (line 365) does not seem to match the plot.

12. Unknown population of cells in single cell RNA sequencing data should further explore. The unique cell population identified in single cell RNA seq may participate in physiological and disease processes.

Reviewers #3-4 (Remarks to the Author):

This work represents a new approach to understanding and characterizing the mechanisms underlying the association between pregestational diabetes and its primary feature, hyperglycemia, and congenital heart defects in humans. Several studies have indicated an increased risk for all CHD in mothers with diabetes, despite widely variable genetic and other environmental risk factors for the individual diseases. The authors use a Streptozosin-induced model for pregestational diabetes in mice to replicate the teratogenic environment, then undertake pooled single-cell sequencing of hearts at two developmental stages to identify differences in cell type proportion and gene expression in exposed vs control embryos. Simple statistical analyses were used to identify features of PGD-exposed embryos, and several pathways and distinct developmental stages in cell differentiation were highlighted as candidates for future study of contributing molecular pathways for human CHD resulting from exposure to maternal hypoglycemia.

Major comments:

1. Please indicate within the methods or figure legends what statistical tests were used for which analyses and in the results please report the p-values for any comparison designated significant (DEseq and cell-fate mapping).
2. The authors should clarify whether the number of analyzed cells per sample is sufficient for detecting differential cell-type gene expression levels using power calculations. With only an average of 500-700 analyzed cells per sample for each timepoint and group there could be a high degree of variation that could skew the proportions (e.g. inflated proportion estimates).
3. The authors do not distinguish between different types of CHD in the introduction or discussion, implying that prevalence, heritability, and known environmental risk factors are similar across all CHD. The authors should dedicate a paragraph in the results to describing the phenotype(s) observed in the embryonic hearts as compared to the wildtype group and discuss relevance to specific human CHD. This clarification will more effectively guide future studies based on this work.
4. The authors do not mention any limitations of their study, whereas it seems increased statistical power and improved capture rate of higher numbers of cells would improve the overall analysis and interpretation of results. For instance, they should note that more samples or cells per samples would be necessary to perform individual level comparisons and also to identify rare cell types or transition states. A paragraph toward the end of the discussion section about which aspects of the methods and results would benefit from cautious interpretation/additional future experimentation would be beneficial.

Minor comments:

1. Title: Since all of the studies were performed in mouse tissues, it would be more accurate to include either "mouse" or "murine model" in the title--while induced hyperglycemia in mice is an excellent model for identifying areas for further characterization in human disease, it cannot fully reproduce the environmental or the genetic factors underlying human CHD.
2. Figures and legends could be clarified to improve interpretation by the reader. Figure 1E should include the mean number of cells per condition under each bar. This is also relevant to several other similar charts in figures 3 and 4.
3. Figure 2B: the color scheme does not add new information and could be easily confused for IF by the reader, thus it is recommended to revert this to a white background, which also applies to FigS8C. In contrast, the colors in Figure 2A are should be changed to avoid confusion in colorblind readers.
4. Figure 5: please label structures in both conditions rather than just the control condition for IF. With regard to labeling figures, multiple asterisks are not necessary for any p-value lower than 0.05 (e.g. Fig 5), as this could misrepresent the overall significance of the finding.
5. Abstract: shorten background, add one more sentence about results/implications
6. Background: "Epidemiological studies have suggested that the subtypes of CHD found in infants exposed to matPGDM range from..." this implies that every one of these conditions have a significantly increased risk with matPGDM, if this is not the case please clarify. The model system and paper focus on maternal hyperglycemia that is described as being similar to T1D, however it seems like the vast majority of pregnant women with diabetes actually have Type 2. The authors should clarify in the discussion whether their model is more relevant to T1D pregnancies or could be applied equally to all pregestational diabetes diagnoses in future human-model studies.
7. Methods: Monocle 2.0 is used in this paper, however the authors should determine whether similar observations could be made with Monocle 3, which was released in February 2019.

8. Methods: For DEseq, this part is unclear (line 676): "combined counts obtained for each gene to produce three in silico replicates of gene vs. expression count matrix." Please expand on whether 'replicates' indicates the same counts were used multiple times, or the cell populations were divided in three by cell type and combined to generate separate but similar replicates.

9. Any time "as previously described" is mentioned in the methods, a citation should be added and a brief description of what was done should still be included.

10. The authors should explicitly state how many hearts were evaluated for each group and stage for the following experiments for each condition: SHF lineage tracing with Isl1 transgenic mice; IF; Tissue dissociation/cell sorting of GFP cells.

11. The authors use an R package developed in their lab entitled "Ryabhatta", as described on their website to perform relevant single-cell analysis functions based on other published and commonly used packages. However the website indicates the package is indexed in the cran repository when it does not appear to be, no github page is listed on the website, and a login is required to view the "download" portion of the authors' website. The availability of this R package should be easily accessible for readers with documentation since it is used extensively throughout the manuscript.

First, the authors thank all the reviewers for their time and providing insightful comments on our manuscript. We have revised the manuscript after addressing the questions, suggestions and concerns from the reviewers, details of which can be found below. We have highlighted the changes in the revised version of our manuscript COMMSBIO-21-3533-T (**highlighted in yellow**).

Reviewer #1:

The authors obtained single-cell RNA-seq expression data of the hearts from maternal hyperglycemia (mathHG) model mice and show that mathHG affects transcriptional dysregulation during heart development and especially disrupts the populations of Isl1+ second heart progenitors and Tnnt2+ cardiomyocytes. Although these findings are interesting, almost all findings in the manuscript are led by single-cell RNA-seq data, which is derived from one mouse for each condition (CNTRL E9.5, CNTRL 11.5, mathHG E9.5, mathHG E11.5). First of all, considering the biological variability, the authors should obtain single-cell data from several organisms for each condition (biological replicates), and assess the biological variability of cell fractions and gene expression levels.

Authors' Response: Thank you very much for reviewing our manuscript and agreeing that these findings are significant. To clarify your comment regarding “single-cell RNA-seq data, which is derived from one mouse for each condition”, we want to emphasize that in the prior version of the manuscript we mentioned use of ≥ 6 embryos/timepoint/maternal diabetic status (described in the methods, results, and figure legends). We apologize for this confusion due to Fig. 1A, which demonstrated a single representative embryo to show the cardiac regions dissected and used for downstream single-cell preparation. Based on our previous experience with murine model of matPGDM, one C57BL/6J female provides 6-8 embryos. *To clarify, $n \geq 6$ embryonic whole hearts (obtained from one litter) in each experimental condition were pooled together, dissociated into single-cell suspension, and used for scRNA-seq library preparation and sequencing (page #6, line 163-167).* While individual embryos were not barcoded *a priori*, we performed *post hoc* sequencing analysis to infer the genotypes of the embryos using additional bioinformatic analysis. According to the reviewer’s comment, we have clearly stated the number of embryonic hearts used for scRNA-seq analysis in the results and methods related to **Fig. 1A** and incorporated additional bioinformatics data to reflect the presence of ≥ 6 embryos/sample (CNTRL E9.5, CNTRL 11.5, mathHG E9.5, mathHG E11.5) using “soupocell” (**Fig. S5A-E**) in the revised manuscript. We have highlighted the changes (in yellow) associated with this change in the revised version of our manuscript and uploaded the genotype data (vcf files for each sample) to the Figshare repository. Since mathHG dose is another important variable and affects the severity of cardiac malformation, we chose to use all the embryos from the same litters rather than pooling from multiple litters.

The findings led by scRNA-seq experiment was further validated using an in vivo reporter mouse model for cell fate mapping studies to examine the impact of mathHG on Isl1+second heart field progenitor cells and cardiomyocytes (**Fig. 4B-P, Fig. 5A-P and Fig. 7A-K**). The potential limitations of this study regarding the sample size have been described in a separate “Limitations” section (**page #23-24, line 665-697**) in the revised manuscript.

Other points.

1. *In Figure 2C-E, the authors conducted GO analysis of DEG. I would like the authors to show the differences in expression levels of representative genes for example using violin plot or heatmap.*

Authors' Response. Thank you for the comment. Figure 2 has now been changed to **Fig. S8A-E** in the revised manuscript. As suggested by the reviewer, we have added a new supplementary figure (**Fig. S9A-C**) to demonstrate the differences in expression levels of representative genes from the GO analysis of DEG in MP and CM clusters.

2. From line 264, the authors describe that given the contribution of MP subpopulations during heart development, these cellular differences suggest a potential role of glucose sensitivity in cells from the SHF and increased risk of CHD in the diabetic offspring. Please cite the relevant papers.

Authors' Response: Thanks for the comment. The basis of this statement relies on the fact that several CHD-causing human variants (in cardiac TFs, epigenetic modifiers and signaling pathways) have been studied in murine heart development and shown to have an SHF origin. A largest Danish nationwide cohort study on infants of type 1 and type 2 matPGDM mothers revealed that the relative risk of CHD originating from the anterior SHF (truncus arteriosus, Tetralogy of Fallot, double-outlet right ventricle, left ventricular outflow tract obstruction, ventricular septum defect) was significantly different from those defects from the posterior SHF. We have added relevant references to the above statement as below in the revised manuscript (**page #11, line 302-304**).

“Given the contribution of MP cell subpopulations in heart development, the changes in proportions of progenitor subpopulations suggest a potential role of glucose sensitivity in cells from the SHF and increased risk of CHD in the diabetic offspring^{16,46,47}.”

3. In Figures 2 and 3, the authors detected the differences in gene expression levels of multipotent progenitors (MP) and cardiomyocytes (CM) between CNTRL and mathHG. The authors should validate these transcriptional changes for example using RNA in situ hybridization.

Authors' Response: Thanks for the suggestion. While RNA ISH dependent validation is an excellent advice, we confirmed the gene expression in MP and CM clusters from scRNA-seq using an independent dual reporter mouse line (*Rosa^{mT/mG}*). Second heart field derived cells (*Isl1-Cre⁺; Rosa^{mT/mG}*) from CNTRL vs. mathHG-exposed E9.5 and E11.5 hearts were obtained similar to scRNA-seq experiment. Cre- animals were used as littermate controls. GFP+ and GFP- cells were sorted from the same animal using flow cytometry and MP and CM-specific gene expressions were validated using SYBR Green based qRT-PCR (**Fig. 5K-P**). The qRT-PCR assay was performed on three pooled embryos/timepoint/maternal condition and on three independent replicates. This data has been included in the revised manuscript (**Figure 5K-P**). The qRT-PCR based gene expression analysis in *Isl1-Cre⁺; Rosa^{mT/mG}* hearts corroborates the transcriptional changes observed in C57BL/6J *wt* hearts from scRNA-seq. Additionally, SHF progenitor markers (*Hand2*, *Nkx2-5*, *Mef2c*, and *Tbx1*) were also validated at the protein level and quantified between CNTRL and mathHG-exposed E9.5 Cre⁺ embryos. This data has been included in the revised manuscript (**Fig. 7A-K and Fig. S14C-G**). The texts have been modified accordingly to reflect these changes and were highlighted.

4. In Figure 4D, the authors used UMAP to reduce dimensions for each cell, but I cannot understand how the authors calculate components 1 and 2 in Figure 4E. This seems strange.

Authors' Response: We recognize that we had not adequately explained the methodology used for our pseudotime analysis and data presentation. In the revised version, Fig. 3 (Figure 4 is earlier version) shows Monocle2 and Slingshot based pseudotime trajectory, where the state IDs are matched. To

avoid the redundancy, we have removed panel D and this figure has been renamed as **Fig. 3A-D**. We have extended our methods section, where we explain the pseudotime analysis in detail (**page #31-32, line 923-934**), as described below.

"We used Ryabhatta (monocle2) to perform pseudotime analysis, which involves a dimensionality reduction step using the DDRTree algorithm. The dimensions of this reduction are called "Component_1" and "Component_2" (analogous to UMAP_1 and UMAP_2). In this case, Component 1 closely tracks "pseudotime" calculated by the "orderCells" function in monocle2. Therefore, we created a new dimensionality reduction using the calculated "pseudotime" as the actual "Component_1" and used the UMAP_1 dimension from RunUMAP function (Seurat) reduction. The goal of this approach is to identify the relative changes of cell types tracked using UMAP_1 with the Component_1 (pseudotime) from monocle2. To estimate the trajectory of the cells along with this dimensionality reduction, we used Slingshot. This analysis permitted us to track the different sub-clusters identified in the MP and CM clusters."

5. In line 337, Fig. 5G should be replaced with Fig. 4G.

Authors' Response: Thank you for identifying this error, we have corrected this in the revised manuscript and the figure number has been changed.

6. From line 352, the authors describe "we found higher number of Isl1+ Tnnt2 + cells at the less differentiated intermediate State 2 in mathHG-exposed E9.5, while the reduction in Tnnt2 + cells was observed in more differentiated CMs at E11.5 State 4 (Fig. 5A)." Please show this quantitatively.

Authors' Response: As suggested by the reviewer, we have added the quantification for Isl1+Tnnt2+ cells at state 2 and Tnnt2+ cells at state 3 in CNTRL and mathHG-exposed E9.5 and E11.5 hearts. Statistical comparisons were made between CNTRL and mathHG-exposed groups using the chi-square test with Yate's correction. The manuscript text, figure legend has been modified accordingly. The figure has been modified as **Fig. 4A** in the revised manuscript.

7. How does mathHG affect the ISL1-associated gene regulatory network? Epigenetic regulation? Please cite some relevant article associated with possible underlying mechanisms in the Discussion section.

Authors' Response: We have previously demonstrated that mathHG alters genome-wide chromatin accessibility, which could lead to differences in gene expression by an in vitro ATAC-sequencing (*ref: 24; Basu et al, JCI Insight 2, 2017, doi: 10.1172/jci.insight.95085*). In this study, mathHG-induced perturbations in Isl1-GRN suggest a probable epigenetic mechanism, which might affect CM differentiation. Previous studies have also shown that LIM-domain-binding protein 1 (Ldb1) is a central regulator of genome organization in cardiac progenitor cells, which is crucial for cardiac lineage differentiation and heart development (*ref: 87; Deng et al; Cell, 149 (2012), pp. 1233-1244*). Another study by Caputo et al; have shown that the Isl1/Ldb1 complex is able to orchestrate genome-wide chromatin organization to instruct the differentiation of multipotent cardiac progenitors (*ref 86: Caputo et al.; Cell Stem Cell. 2015 Sep 3;17(3):287-99*). Therefore, we hypothesize that mathHG exposure impacts long-range chromatin interaction(s) with Isl1-interacting loci and results in gene dysregulation. As suggested by the reviewer, we have cited relevant articles associated with possible underlying mechanisms in the Discussion section of the revised manuscript (**page #21, line 607-611**).

8. In line 365, Fig. 8A should be replaced with Fig. S8A.

Authors Response: We have corrected this in the revised manuscript and the figure number has been changed. Thank you!

Reviewer #2 (Remarks to the Author):

This is an interesting study using some state-of-the-art technologies. Manivannan et al. reported the embryonic response to mathHG environment, especially *Isl1*+ second heart field progenitors and *Tnnt2*+ cardiomyocytes. Then through *Isl1-Cre^{+/-}* and *RosamT/mG* dual reporter mice, the authors confirmed that mathHG exposure leads to impaired cardiomyocyte differentiation. It has a well-thought logical structure, and the figures and tables overall are reluctantly supporting the premise of the study. However, there are some major concerns that arise from looking at the experimental data that need unambiguous clarifications or thorough revision before possible publication.

Major concerns:

1. There was only 1 sample at each group, which makes it very difficult to evaluate the differences among samples. This also probably led to accidental error. Therefore, the authors should increase sample size for single cell RNA sequencing if possible.

Authors' Response: First of all, thank you very much for appreciating our findings and considering this to be relevant to the field of teratogen induced risk of CHD. We recognize that we had not adequately explained the methodology used for single-cell RNA-sequencing. We used the entire litter per sample, where each sample meant cells were obtained from pooling at least 6 embryos/timepoint exposed to control and maternal hyperglycemic environment. This has been mentioned in our previous version in the methods, results, and figure legends. The schematics in **Fig. 1A** only demonstrated a single representative embryo to show the cardiac regions dissected and used for downstream analysis. The embryonic hearts at each experimental condition were pooled, dissociated, and used for scRNA-seq library preparation and next-generation sequencing. However, this time we have clarified the number of embryonic hearts used for scRNA-seq analysis by adding this information in results section associated with Fig. 1A (**page #6, line 163-167**). In addition, we have performed SNP genotyping using “souporecell” to reflect the presence of ≥ 6 embryos/sample (CNTRL E9.5, CNTRL 11.5, mathHG E9.5, mathHG E11.5). We have highlighted these changes in the revised version of our manuscript (**Fig. S5A-E**) and uploaded the genotype data of each sample to Figshare repository. Since mathHG dose is another important variable and affects the severity of cardiac malformation, we chose to use all the embryos from the same litters rather than pooling from multiple litters.

2. I addressed all of my concerns regarding data processing. Nearly half of the single cell RNA sequencing data was discarded (Line 169-171), which could result in pronounced changes to cell constitution and accuracy of follow-up data analysis. Thus, the authors should detailly explain the motivations behind the data processing, and they also should include analyses to show that alternative processing of the data give rise to the same conclusions.

Authors' Response: Thank you for the comment. The stringent QC steps adopted in this study follow previously published scRNA-seq data processing methods in embryonic heart tissues (**ref: 43: de**

Soysa TY. et al., *Nature*. 2019 Aug;572 (7767):120-124, **ref: 79**, DeLaughter DM, et al; *Dev Cell*. 2016 Nov 21;39(4):480-490). Cells showing $\geq 10\%$ mitochondrial content, low UMI count, endodermal, ectodermal, and blood/immune cells were discarded for further analysis. We have provided a detailed flow chart in **Fig. S1** explaining the data processing steps and highlighted the text to reflect this change.

In addition, we have extended the analysis by integrating our data with de Soysa TY. et al., *Nature*. 2019 Aug;572(7767):120-124 control E9.25 mouse heart cells (**Fig. S2H**). This integration shows that the cell types annotated by us using cluster markers identified in our data closely track the cell types annotated by de Soysa TY. et al., 2019. Moreover, clusters that we removed from our analysis for downstream examination did not have a pair in deSoysa TY. et al., 2019. These clusters include clusters showing ectodermal markers, low UMI counts, and immune and blood cell markers. As described in de Soysa et al., similar clusters were also identified in their raw data (de Soysa TY. et al., 2019 Extended Figure data 1), but excluded for downstream analysis to study early cardiac development. We have highlighted these changes in the revised version of our manuscript (**page #7, line 186-193 in Results section and page #28-29, line 809-852**)

3. In this study, the authors keep only the cells of mesodermal and mesectodermal origin in further analysis. However, there remain endocardial/endothelial (EC) and epicardial (EP) in cluster definition, how to explain it?

Authors' Response: The authors appreciate your query. In this study, the rationale for using single-cell RNA-sequencing in developing embryonic hearts was to demonstrate cell-type-specific transcriptional changes in response to matHG. Since matHG is a potent teratogen, it affects a lot of cell types in utero. To align with the clinical manifestation of maternal diabetes induced conotruncal and septal defects which clearly suggest an SHF origin, we primarily focused on studying differentially expressed genes in multipotent progenitor and cardiomyocyte (MP-CM) clusters and compared between groups. That being said, we have not excluded the contribution of endocardial/endothelial (EC), fibromesenchymal (FM), epicardial (EP) and cardiac neural crest cells (NC) from our analysis. The DEG analysis in these clusters were shown in **Fig. S16A-I and Figure S17A-G**, respectively. Earlier studies in murine model of matHG have already shown dysregulation of gene expression in EC, NC that led to endocardial/endothelial to mesenchymal transition (EMT) and neural crest migration defects linked to matPGDM associated CHD. But their cell lineage specific role in inducing CHD is beyond the scope of this work. We have added this to the limitation section of the revised manuscript (**page #23-24, line 665-697**).

4. As for the comparisons of the cellular proportions, rejecting large amounts of data and insufficient sample sizes make a low accuracy. Those results should be confirmed by other experiments.

Authors' Response: We respectfully disagree with this comment about accuracy owing to the following reasons:

- 1) As mentioned earlier, we have pooled at least 6 hearts to compare gene expression analysis between CNTRL and matHG-exposed E9.5 and E11.5 embryos. We have confirmed the presence of at least 6 distinguishable genotypes per sample per timepoint, using *post hoc* sequencing evaluation of SNPs using souporecell. We have a suitable number of biological replicates to test our hypothesis.
- 2) In response to the reviewer's comments, we performed the power analysis on the single-cell data using two separate methods a) Using SCOPIT to calculate the number of cells required to capture low abundant cell types. We show that we could detect a cell population with a frequency

of 1% in all the cells can be detected with the number of cells we have captured per sample per time point with high confidence. b) Post-hoc analysis using SCOPIT to show that the cell types with the least frequency in each sample can be detected in other samples with high confidence using the number of cells captured by our single-cell data. This suggests that our sample size (in terms of the number of cells) is sufficient to capture every cell type observed in control data across test conditions. This also means that we are controlling for sampling bias when evaluating cellular proportions. This analysis is now included in our revised manuscript (**Fig. S4A-C**).

- 3) We have also tabulated the number of cells (non-empty) droplets identified by 10x Genomics Cell Ranger and the number of cells retained after our quality control analysis. This shows that we have indeed retained most of the cells suitable for the downstream analysis.
- 4) The cell numbers at the abovementioned time points (obtained from our study) mirror those observed in previously published scRNA-seq data at similar developmental timepoints (*de Soysa TY. et al., Nature. 2019 Aug;572(7767):120-124, DeLaughter DM, et al; Dev Cell. 2016 Nov 21;39(4):480-490, Guang L, Dev Cell. 2016 Nov 21; 39(4): 491–507*). We have extended our analysis to demonstrate that we detected similar cellular distribution as shown in *deSoysa TY. et al., Nature. 2019 Aug;572(7767):120-124* (**Fig S2H**).
- 5) Additionally, we have added the detailed data processing steps (shown as a flow chart in **Fig. S1**) in the revised manuscript. This pipeline justifies the removal of ectodermal, endodermal, low UMI, and high-mitochondrial content cells from further analysis.
- 6) To further strengthen our observations and results, we validated the transcriptional changes in an independent set of embryos (obtained from independent dams) at both RNA (qRT-PCR) and protein (immunostaining) levels.
- 7) Finally, we have performed *in vivo* cell-lineage tracing experiments to specifically demonstrate that *matHG* impedes second heart field-derived cardiomyocyte differentiation and proliferation (**Fig. 4B-P, Fig. 5A-P and Fig. 7A-K**). Future genetic cell fate mapping studies targeting other cardiac cell types will unravel the mechanism of specific CHDs observed in the offspring of diabetic mothers.

5. Line 257-259, the CMs further divided into outflow tract (OFT), atrioventricular canal (AVC), atrial (Atr) and ventricular (Ven)-CM subpopulations. What basis is there for this?

Authors' Response: Thank you for the comment. In normal cardiogenesis, the initial cardiac crescent structure representing cells from the first and second heart fields (FHF and SHF), cardiac precursors undergo midline fusion to generate a linear heart tube that undergoes looping morphogenesis, which represents the first asymmetric morphologic event during development (*Bruneau, Nature. 2008; 451:943–948.; Buckingham et al., Nat Rev Genet. 2005; 6:826–835*). Subsequent ballooning and expansion of chamber-specific cardiomyocytes, followed by delamination and trabeculation, produce clear distinctions between the atria (Atr), ventricles (Ven), outflow tract (OFT), and the atrioventricular canal (AVC) as described in *Guang L, Dev Cell. 2016 Nov 21; 39(4): 491–507*. In this study, we obtained scRNA-seq data in embryos post looping morphogenesis. The basis for the sub-clustering CM was to examine how *matHG* affects gene expression in cardiomyocyte subtypes present in the embryonic heart at E9.5 and E11.5. We added one sentence in the Discussion section of the revised manuscript to clarify the rationale behind the sub-clustering of the CM to OFT-CM, AVC-CM, Atr-CM, and Ven-CM (**page #23-24, line 665-697**) as described below:

“The cardiomyocyte is the fundamental work unit of the heart. Although CM fate is a developmental end point, a diversity of CM subtypes exists within the heart⁸³. After ballooning morphogenesis, the

expansion of chamber-specific CM is followed by delamination and trabeculation, which produce clear distinctions between the Atr-CM, Ven-CM, OFT-CM, and AVC-CM⁴².

Minor concerns:

1. The authors should add relevant references about CHD in line 63-65.

Authors' Response: We have added relevant references (*ref 1, 2 in the revised manuscript*.)

2. Line 74 “increased occurrence of CHD”, consider revising to “increased occurrence of CHD in fetuses”

Authors' Response: We have revised the above sentence to “increased occurrence of CHD in the infants of diabetic mothers” (*page #3, line 72*).

3. Line 81~82. I wonder if “transposition of great arteries, outflow tract (OFT) defects with normally related great arteries, double outlet right ventricle (DORV)” could be simplified as “conotruncal defects”

Authors' Response: The authors thank you for the suggestion. Since conotruncal defects are thought to result from a disturbance of the outflow tract of the embryonic heart, and comprise truncus arteriosus, tetralogy of Fallot, interrupted aortic arch, transposition of the great arteries, and double outlet right ventricle, we simplified the sentence by modifying it to “conotruncal defects”.

4. Please delete the “and their derivatives” in Line 100.

Authors' Response: Thank you for your suggestion. It has been deleted in the revised manuscript.

5. “*****and when deleted in mice demonstrate embryonic lethality and higher incidences of CHD”, please check and correct it

Authors' Response: We have corrected the sentence. Thank you for pointing this out.

6. In terms of language and style, the manuscript itself needs to make improvements. There are too many individual comments to list them all here.

Authors' Response: The manuscript has been edited and proofread by Dr. Dennis Lewandowski in the Heart Center Editorial Service Core at Nationwide Children’s Hospital.

7. I suggest that authors include a paragraph listing the limitation of this study including small sample size

Authors' Response: We have included a separate paragraph describing the potential limitations of this study (*page #23-24, line 665-697*).

8. In part “Maternal hyperglycemia triggers key molecular changes in cardiac progenitors and cardiomyocytes”, what is the experimental group and control group in each differential expression analysis should be clearly described to avoid any ambiguity.

Authors' Response: We have clearly described the experimental (mathG-exposed embryos) and control (CNTRL or normoglycemia exposed embryos) groups in differential gene expression analysis in the Methods, Results, and Figure legends of the revised manuscript.

9. Line 270-272 “Atr and Ven-CMs significantly higher in the E11.5 hearts”, while Line 279-281 “the expression of Gja1+ Ven-CMs, Bmp4+, Rgs5+, and Dlk1+ OFT-CMs, Cacna2d2+, Tbx3+ AVC-CMs and Kcan5+, Nr2f1+, Nr2f2+ Atr-CMs were reduced with mathG”. Please check this inconsistency.

Authors' Response: We have checked for inconsistency in the sentence mentioned above and revised it. To clarify the underlying basis of differences in cellular distribution, we have described the significant upregulation of genes in mathG-exposed CM subtypes compared to CNTRL embryos at E11.5, in the revised manuscript (**page #11, line 317-324**).

10. Figure 4D is quite confused for me. From this plot, each subpopulation seems to be in a unique state, this does not agree with Figure 4B.

Authors' Response: We apologize for the confusion. Figure 4B describes the cellular distribution in five pseudotime states derived from Monocle 2. Figure 4D shows the merged UMAP plot of MP-CM subpopulations, which has been described in new Figure 3. Due to redundancy, we have removed Figure 4D in the revised version of the manuscript and the figure number has been changed to **Fig. 3A-D** in the revised manuscript.

11. The Figure 8A (line 365) does not seem to match the plot.

Authors' Response: We have corrected the Figure numbers and matched them with the respective plot in the revised manuscript.

12. Unknown population of cells in single cell RNA sequencing data should further explore. The unique cell population identified in single cell RNA seq may participate in physiological and disease processes.

Authors' Response: We have added a section in Discussion (**page #22, line 658-663**) stating that *“The unknown population of the cells (cluster 11) was found to have primarily immune cell markers. However, there were additional markers suggesting that this cluster might constitute doublets or multiplets. To avoid an ambiguous name (immune/doublet) we decided to leave these cells as unknown. Analysis of this cell type, if not doublets, would be interesting from an immune response standpoint. However, it is beyond the scope of this study”*.

The Limitation (**page #23-24, line 665-697**) section also describes that *the ability to understand the complexity of a cell population requires careful analysis from all aspects*.

Reviewers #3-4 (Remarks to the Author):

This work represents a new approach to understanding and characterizing the mechanisms underlying the association between pregestational diabetes and its primary feature, hyperglycemia, and congenital heart defects in humans. Several studies have indicated an increased risk for all CHD in mothers with diabetes, despite widely variable genetic and other environmental risk factors for the individual diseases. The authors use a Streptozotocin-induced model for pregestational diabetes in mice to replicate the teratogenic environment, then undertake pooled single-cell sequencing of hearts at two

developmental stages to identify differences in cell type proportion and gene expression in exposed vs control embryos. Simple statistical analyses were used to identify features of PGD-exposed embryos, and several pathways and distinct developmental stages in cell differentiation were highlighted as candidates for future study of contributing molecular pathways for human CHD resulting from exposure to maternal hypoglycemia.

Thank you very much for appreciating the novelty of our findings. As you correctly mentioned, this cardiac cell type specific transcriptional responses in maternal hyperglycemia exposed embryos will facilitate our understanding about the mechanistic basis of CHD in diabetic pregnancies.

Major comments:

1. Please indicate within the methods or figure legends what statistical tests were used for which analyses and, in the results, please report the p-values for any comparison designated significant (DEseq and cell-fate mapping).

Authors' Response: Thank you for the comment. We have incorporated all the statistical tests used for DESeq2 comparison, cell fate mapping analysis, immunofluorescence quantifications between CNTRL and matHG-exposed embryos within the methods and figure legends. The texts with these changes have been highlighted (in yellow) in the revised manuscript.

2. The authors should clarify whether the number of analyzed cells per sample is sufficient for detecting differential cell-type gene expression levels using power calculations. With only an average of 500-700 analyzed cells per sample for each time point and group there could be a high degree of variation that could skew the proportions (e.g., inflated proportion estimates).

Authors' Response: We recognize that the number of cells in each condition and sub-cluster is not provided in raw number format. To address this comment,

- 1) we performed the power analysis on the single-cell data using two separate methods a) Using SCOPIT to calculate the number of cells required to capture low abundant cell types. We show that we could detect a cell population with a frequency of 1% in all the cells can be detected with the number of cells we have captured per sample per time point with high confidence. b) Post-hoc analysis using SCOPIT to show that the cell types with the least frequency in each sample can be detected in other samples with high confidence using the number of cells captured by our single-cell data. This suggests that our sample size (in terms of the number of cells) is sufficient to capture every cell type observed in control data across test conditions. This also means that we are controlling for sampling bias when evaluating cellular proportions. This analysis is now included in our revised manuscript (**Fig S4A-C**).
- 2) We have also tabulated the number of cells (non-empty) droplets identified by 10x Genomics Cell Ranger and the number of cells retained after our quality control analysis. This shows that we have indeed retained most of the cells suitable for the downstream analysis.

We have added Supplemental **Fig. S4A-C** describing the power analysis in the revised manuscript.

3. The authors do not distinguish between different types of CHD in the introduction or discussion, implying that prevalence, heritability, and known environmental risk factors are similar across all CHD. The authors should dedicate a paragraph in the results to describing the phenotype(s) observed in the

embryonic hearts as compared to the wildtype group and discuss relevance to specific human CHD. This clarification will more effectively guide future studies based on this work.

Authors Response: Thank you very much for your insightful comment. We have added few sentences in the Introduction (**page #3, line 68-75 and line 86-90**) and *Discussion* sections (**page #19, line 541-545**), mentioning the different types of CHD associated with diabetic pregnancies. We also incorporated a paragraph in the *Results* (**page #6, line 152-154**) to describe the phenotype(s) observed in the embryonic hearts as compared to the wildtype group and discussed relevance to specific human CHD. These sections are highlighted in yellow.

4. The authors do not mention any limitations of their study, whereas it seems increased statistical power and improved capture rate of higher numbers of cells would improve the overall analysis and interpretation of results. For instance, they should note that more samples or cells per samples would be necessary to perform individual level comparisons and also to identify rare cell types or transition states. A paragraph toward the end of the discussion section about which aspects of the methods and results would benefit from cautious interpretation/additional future experimentation would be beneficial.

Authors Response: Thank you for the comment. We have added a separate paragraph in the *Discussion* section mentioning about the potential limitations of our study and the requirement for additional in vivo experiments in the revised manuscript (**page #23-24, line 665-697**).

Minor comments:

1. Title: Since all of the studies were performed in mouse tissues, it would be more accurate to include either "mouse" or "murine model" in the title--while induced hyperglycemia in mice is an excellent model for identifying areas for further characterization in human disease, it cannot fully reproduce the environmental or the genetic factors underlying human CHD.

Authors' Response: Thank you for the insightful comment. We have modified the title of the revised manuscript.

The title has been revised to ***Single-cell Transcriptomic Profiling Unveils Dysregulation of Cardiac Progenitor Cells and Cardiomyocytes in Murine Model of Maternal Hyperglycemia to Elevate the Risk of Congenital Heart Defects***, which is more reflective of our findings.

2. Figures and legends could be clarified to improve interpretation by the reader. Figure 1E should include the mean number of cells per condition under each bar. This is also relevant to several other similar charts in figures 3 and 4.

Authors' Response: Thank you! We have clarified the figures and figure legends accordingly. The number of cells per condition per cluster has been included under each bar in the revised manuscript.

3. Figure 2B: the color scheme does not add new information and could be easily confused for IF by the reader, thus it is recommended to revert this to a white background, which also applies to FigS8C. In contrast, the colors in Figure 2A should be changed to avoid confusion in colorblind readers.

Authors' Response: We agree with your suggestion. The color scheme for the ExpressO plots (**in Fig. S8B, S13C and S14A, B**) has been changed to a white background to avoid confusion from IF images.

4. Figure 5: please label structures in both conditions rather than just the control condition for IF. With regard to labeling figures, multiple asterisks are not necessary for any p-value lower than 0.05 (e.g., Fig 5), as this could misrepresent the overall significance of the finding.

Authors Response: Thank you for the suggestion. We have labeled heart structures in both conditions (CNTRL and matHG-exposed embryos) for the IF images. We have removed multiple asterisks from the graphs and added * to signify two-tailed $p < 0.05$ (between group comparisons) in the revised manuscript.

5. Abstract: shorten background, add one more sentence about results/implications

Authors Response: We have modified the abstract as suggested by the reviewer (*page #2, line 33-52*). Thank you!

6. Background: "Epidemiological studies have suggested that the subtypes of CHD found in infants exposed to matPGDM range from..." this implies that every one of these conditions have a significantly increased risk with matPGDM, if this is not the case please clarify. The model system and paper focus on maternal hyperglycemia that is described as being similar to T1D, however it seems like the vast majority of pregnant women with diabetes actually have Type 2. The authors should clarify in the discussion whether their model is more relevant to T1D pregnancies or could be applied equally to all pregestational diabetes diagnoses in future human-model studies.

Authors Response: Thank you for the insightful comment. We have rephrased the statement as following: "In a study published in the American Journal of Preventive Medicine, the researchers found that uncontrolled blood sugar in women with type 1 or type 2 diabetes before pregnancy led to ~2,670 babies with CHD each year^{15,16} and cited the reference (*page #3, line 72-75*). In this study, we emphasized on the role of maternal hyperglycemia, which is the key teratogen in both T1D and T2D pathogenesis. In the *Discussion* section of revised manuscript (highlighted in yellow, *page #19, line 541-545*). we have added a sentence clarifying that although chemically (streptozotocin) induced animal model of maternal pregestational diabetes strongly mimics human T1D pathology, the effect of matHG could be applied equally to all pregestational diabetes diagnoses in future human-model studies (*page #19, line 550-553*).

7. Methods: Monocle 2.0 is used in this paper; however, the authors should determine whether similar observations could be made with Monocle 3, which was released in February 2019.

Authors Response: We appreciate your comment. The two versions of the R package monocle2 and monocle3 are similar in providing pseudotime analysis, except monocle3 does not support DDRTree algorithm according to Cole Trapnell (<https://github.com/cole-trapnell-lab/monocle3/issues/187#issuecomment-540229295>). This algorithm while similar to SimplePPT used in monocle3 provides a robust dimensionality reduction that closely tracks the pseudotime calculated from the ordered cells. Additionally, monocle 2 is distributed through CRAN (the major R-package repository) and is compatible with several package versions used in the app. Monocle 3 is not available through CRAN or Bioconductor and requires versions of packages that conflict with the others used by R packages used in Ryabhatta. For these reasons we have used Monocle 2.0 in our analysis. To further

support our trajectory analysis we have used Slingshot, which provides a similar minimum spanning tree structure for the cells as observed using monocle.

8. Methods: For DEseq, this part is unclear (line 676): "combined counts obtained for each gene to produce three in silico replicates of gene vs. expression count matrix." Please expand on whether 'replicates' indicates the same counts were used multiple times, or the cell populations were divided in three by cell type and combined to generate separate but similar replicates.

Authors' Response: Thank you for your comment. As described in Ryabhatta's pre-print (<https://www.biorxiv.org/content/10.1101/2021.06.17.448424v1>), the differential expression is achieved by virtually splitting the cells in each cluster into four groups. These four groups are then used as replicates. The raw counts from each of the replicates are then added along each gene to provide a pseudo-bulk count from a virtual replicate. The 3 virtual replicates per sample (maternal diabetes status and time point) per cluster is then used to perform DESeq2-based differential expression analysis. The cells from one group (or replicate) are not included or repeated in any other group.

9. Any time "as previously described" is mentioned in the methods, a citation should be added and a brief description of what was done should still be included.

Authors' Response: Thank you for the comment. We have added citation(s) and short description of previously described methods in the revised manuscript, whenever necessary.

10. The authors should explicitly state how many hearts were evaluated for each group and stage for the following experiments for each condition: SHF lineage tracing with Isl1 transgenic mice; IF; Tissue dissociation/cell sorting of GFP cells.

Authors' Response: Thank you for the comment. We have clearly described the number of embryonic hearts used for in vivo lineage tracing, tissue dissociation, and cell sorting experiments in the *Methods and Figure Legends* of the revised manuscript.

11. The authors use an R package developed in their lab entitled "Ryabhatta", as described on their website to perform relevant single-cell analysis functions based on other published and commonly used packages. However, the website indicates the package is indexed in the cran repository when it does not appear to be, no GitHub page is listed on the website, and a login is required to view the "download" portion of the authors' website. The availability of this R package should be easily accessible for readers with documentation since it is used extensively throughout the manuscript.

Authors' Response: We appreciate the interest in the app and as observed on the website, we have made it available free of cost. The registration page is there to support us track the number of unique users. This information is necessary to support future funding for the app development. Please note that, GitHub and Gitlab do not provide the ability to track downloads (only forking requests), Zotero and other apps document the number of downloads and not unique users.

Since our target users are non-computational biologists, we find providing the tool through a direct download page is the best option. We do not store any private information. An algorithm checks if the provided email exists in the list of existing users and adds a new user when the email is new. We do not send any verification requests or spam the users with emails. The website does not mention that the package is indexed in CRAN. It provides a link to the cran repository to download and install for

users who wish to run the script through RStudio. The script file can be downloaded from the website. Alternatively, a no-install version has been developed and made available through the website. This version of the app runs on Mac and Windows computers without the need to install R and RStudio. We have also compiled a free (open access) manual for all users using the following link: <https://natian-and-ryabhatta.web.app/>

REVIEWERS' COMMENTS:

Reviewer #1 (Remarks to the Author):

The authors have appropriately responded to my concerns.

Reviewers #3-4 (Remarks to the Author):

Thank you for addressing each of these comments and improving the manuscript. I have no further questions or concerns.